# CONTINUOUS-TIME IDENTIFICATION OF DYNAMIC STATE-SPACE MODELS BY DEEP SUBSPACE ENCODING

**Gerben I. Beintema, Maarten Schoukens & Roland Tóth**[*]
Department of Electrical Engineering, Eindhoven University of Technology, The Netherlands
{g.i.beintema,m.schoukens,r.toth}@tue.nl

## ABSTRACT

Continuous-time (CT) modeling has proven to provide improved sample efficiency and interpretability in learning the dynamical behavior of physical systems compared to discrete-time (DT) models. However, even with numerous recent developments, the CT nonlinear state-space (NL-SS) model identification problem remains to be solved in full, considering common experimental aspects such as the presence of external inputs, measurement noise, latent states, and general robustness. This paper presents a novel estimation method that addresses all these aspects and that can obtain state-of-the-art results on multiple benchmarks with compact fully connected neural networks capturing the CT dynamics. The proposed estimation method called the subspace encoder approach (SUBNET) ascertains these results by efficiently approximating the complete simulation loss by evaluating short simulations on subsections of the data, by using an encoder function to estimate the initial state for each subsection and a novel state-derivative normalization to ensure stability and good numerical conditioning of the training process. We prove that the use of subsections increases cost function smoothness together with the necessary requirements for the existence of the encoder function and we show that the proposed state-derivative normalization is essential for reliable estimation of CT NL-SS models.

## 1    INTRODUCTION

Dynamical systems described by nonlinear state-space models with a state vector $x(t) \in \mathbb{R}^{n_x}$ are powerful tools of many modern sciences and engineering disciplines to understand potentially complex dynamical systems. One can distinguish between Discrete-Time (DT) $x_{k+1} = f(x_k, u_k)$ and Continuous-Time (CT) $\frac{dx(t)}{dt} = f(x(t), u(t))$ state-space models. In general, obtaining DT dynamical models from data is easier than CT models since data in computers is represented as discrete elements (e.g. arrays). However, the additional implementation complexity and computational costs associated with identifying CT models can be justified in many cases. First and foremost, from the natural sciences, we know that many systems are compactly described by CT dynamics which makes the continuity prior of CT models a well-motivated regularization/prior (De Brouwer et al., 2019). It has been observed that this regularization can be beneficial for sample efficiency (De Brouwer et al., 2019) which is a common observation when "including physics" in learning approaches (Karniadakis et al., 2021). Furthermore, the analysis of ODE equations is a well-regarded field of study with many powerful results and methods which could further improve model interpretability (Fan et al., 2021), such as applied in Bai et al. (2019). Another inherent advantage is that these models can be used with irregularly sampled or missing data (Rudy et al., 2019). Lastly, in the control community, CT models are generally regarded desirable for control synthesis tasks as shaping the behavior of the controller is much more intuitive in CT (Garcia et al., 1989). Hence, developing robust and general CT models and estimation methods would be greatly beneficial.

In the identification of physical CT systems, it is common to encounter challenges such as: external inputs ($u(t)$), noisy measurements, latent states, unknown measurement function/distribution (e.g. $y(t) = h(x(t))$), the need for accurate long-term predictions and a need for a sufficiently low computational cost. For instance, all these aspects need to be considered for the cascade tank benchmark

---

[*]Also associated with, Systems and Control Laboratory, Institute for Computer Science and Control, Budapest, Hungary.

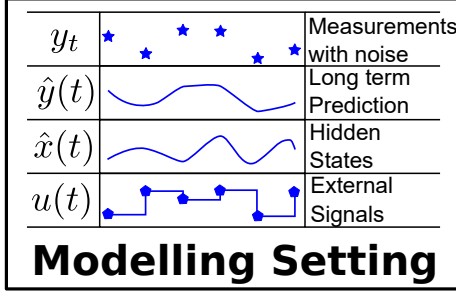 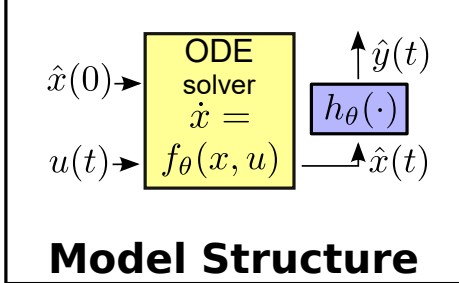

**Figure 1:** In this work, we consider the problem of estimating continuous-time (CT) state-space models from noisy observation (additive noise) with long-term prediction capabilities, hidden states and external signals in a computationally efficient and robust manner.

problem (Schoukens & Noël, 2017). These aspects and the considered CT state-space model is summarized in Figure 1. Many of these aspects have been studied independently, for instance, Brajard et al. (2020); Rudy et al. (2019) explicitly addressed the presence of noise on the measurement data, Maulik et al. (2020); Chen et al. (2018) provided methods for modeling dynamics with latent states, Zhong et al. (2020) considers the presence of known external inputs, Zhou et al. (2021a) provides a computationally tractable method for accurate long-term sequence modeling. However, formulating models and estimation methods for the combination of multiple or all aspects is in comparison underdeveloped with only a few attempts such as Forgione & Piga (2021a) that have been made.

In contrast to previous work, we present a CT encoder-based method which is a general, robust and well-performing estimation method for CT state-space model identification. That is, the formulation addresses noise assumptions, external inputs, latent states, an unknown output function, and provides state-of-the-art results on multiple benchmarks of real systems. The presented subspace encoder method is summarized in Figure 2. The proposed method considers a cost function evaluations on only short subsections of the available dataset which reduces the computational complexity. Furthermore, we show theoretically that considering subsections enhances cost function smoothness and thus optimization stability. The initial states of these subsections are estimated using the encoder function for which we present necessary requirements for its existence. Lastly, we introduce a normalization of the state and state-derivative and we show that it is required for proper CT estimation. Moreover, we attain additional novelty as these results are obtained without needing to impose a specific structure on the state-space (such as in Greydanus et al. (2019); Cranmer et al. (2020)) obtaining a practically widely applicable method.

Our main contributions are the following;

- We formally derive the problem of CT state-space model estimation with latent states, external inputs, and measurement noise.

- We reduce the computational loads by proposing a subspace encoder-based identification algorithm that employs short subsections, an encoder function that estimates the initial latent states of these subsections, and a state-derivative normalization term for robustness.

- We make multiple theoretical contributions; *(i)* we prove that the use of short subsections increases cost function smoothness by Lipschitz continuity analysis, *(ii)* we derive necessary conditions for the encoder function to exist and *(iii)* we show that a state-derivative normalization term is required for proper CT model estimation.

- We demonstrate that the proposed estimation method obtains state-of-the-art results on multiple benchmarks.

## 2 RELATED WORK

One of the most influential papers in CT model estimation is the introduction of neural ODEs (Chen et al., 2018), which showed that residual networks is presented as an Euler discretization of a continuous in-depth neural network. Moreover, they also show that one can employ numerical integrators

to integrate through the depth in a computationally efficient manner. This depth can be interpreted as the time direction to be able to model dynamical systems. The ideas in the neural ODE contribution have been extended to/used in, for instance, normalizing flows to efficiently model arbitrary probability distributions (Papamakarios et al., 2021; Grathwohl et al., 2019), and enhance the understanding and interpretability of neural networks (Fan et al., 2021). However, the neural ODE does not scale well for long sequences, nor does it consider external inputs or noise, and the optimization process is often unstable.

An adjacent research direction is the method/models which consider CT dynamics and directly use the state derivatives and even often the noiseless states to formulate structured and interpretable models such as Hamiltonian Neural Networks (HNN) (Greydanus et al., 2019), Lagrangian Neural Networks (LNN) (Cranmer et al., 2020) and Sparse Identification of Nonlinear Dynamics (SINDy) (Brunton et al., 2016). In contrast, the proposed method is formulated for an unstructured state-space and does not require the system state or the state derivatives to be known.

Our method is in part related to (Ayed et al., 2019) which concerns the estimation of CT models with latent variables. They also employ an encoder function to estimate initial states, however, this encoder is only dependent on the past outputs, contains a partially known state and there is no theoretical support for the method. Furthermore, in that work, only a fixed output function is considered and the involved optimization problem is solved as an optimal control problem whereas our formulation alters the simulation loss function to obtain a computationally desirable form. Furthermore, (Forgione & Piga, 2021a), to which we compare in this work, considers CT model with latent variables, subsections, and an additional loss term for the integration error. However, they include the initial states of these subsections as free optimization parameters. This increases the model complexity with the number of subsections. In contrast, our proposed method uses an encoder to estimate the initial states. This results in fixed model complexity. Furthermore, we only employ a single loss function and a novel state-derivative normalization term. Additionally, we provide theoretical insights into these existing elements and extend them to the considered setting in a robust and computationally efficient manner.

## 3 PROBLEM STATEMENT

Consider a system represented by a continuous-time nonlinear state-space (CT NL-SS) description sampled at a fixed interval $\Delta t$ for simplicity:

$$
\begin{aligned}
\dot{x}_s(t) &= f(x_s(t), u(t)), \\
y_k &= h(x_{s,k}, u_k) + w_k,
\end{aligned}
\tag{1}
$$

where the subscript notation denotes sampling as $x_{s,k} = x_s(k\Delta t)$, $x_s(t) \in \mathbb{R}^{n_{x_s}}$ is the system state variable, $u(t) \in \mathbb{R}^{n_u}$ is the input, $y_k \in \mathbb{R}^{n_y}$ is the output, $f$ represents the system dynamics and $h$ gives the output function while $w_k \in \mathbb{R}^{n_y}$ is a i.i.d. zero-mean white noise process with finite variance $\Sigma_w$.

For this system, the CT model estimation problem can be expressed for a given dataset of measurements:

$$
D_N = \{(u_0, y_0), (u_1, y_1), ..., (u_{N-1}, y_{N-1})\},
$$

with unknown $w_k$, $x_s(t)$, $\dot{x}_s(t)$, $\dot{y}_k$ and initial state $x_s(0)$, as the following optimization problem (a.k.a. simulation loss minimization):

$$
\begin{aligned}
\min_{\theta, x(0)} \quad & \frac{1}{N} \sum_{k=0}^{N-1} \|y(k\Delta t) - \hat{y}(k\Delta t)\|_2^2, \\
\text{s.t.} \quad & \hat{y}(t) = h_\theta(x(t)), \\
& \dot{x}(t) = f_\theta(x(t), u(t)),
\end{aligned}
\tag{2}
$$

where $x(t) \in \mathbb{R}^{n_x}$ is the model state, $h_\theta$ and $f_\theta$ are the output and state-derivative functions parameterized by $\theta$ and being Lipschitz continuous in their inputs and parameterization. These two functions are formulated as multi-layer feedforward neural networks during our experiments.

To obtain the simulation output $\hat{y}(k\Delta t)$, one can integrate $\dot{x}(t) = f_\theta(x(t), u(t))$ starting from the initial state $x(0)$. This integration can be performed with any ODE solver that allows for backpropagation such as Euler ($x(t + \Delta t) = x(t) + \Delta t f_\theta(x(t), u(t))$), RK4, or numerous adaptive step

methods (Chen et al., 2018; Ribeiro et al., 2020).[1] To make this a well-posed optimization problem, additional information or an assumption on the inter-sample behavior of $u(t)$ is required, since, for example, $u(\Delta t/2)$ is not present in $D_N$. This behavior is often chosen to be Zero-Order Hold (ZOH) (Ljung, 1999) as can be viewed in Figure 1.

Multiple major issues are encountered when solving the optimization Problem (2) with a gradient-descent-based method. The first issue is that computing the value of the loss function requires a forward pass on the whole length of the dataset (Ayed et al., 2019). Hence, the computational complexity grows linearly with the length of the dataset. Furthermore, a common occurrence is that the values of $x(t)$ or its gradient grows exponentially which results in non-smooth loss functions or gradients. This causes gradient-based optimization algorithms to become unreliable since the optimization process might be unstable or it converges to a local minima (Ribeiro et al., 2020). All these issues are addressed in the proposed method.

## 4 PROPOSED METHOD

We propose to consider multiple overlapping short subsections of length $T\Delta t$ to form a truncated simulation loss instead of simulating over the entire length of the dataset. We express this in the following optimization problem (note that we express the optimization problem with discrete-time notation ($u_k := u(k\Delta t)$) for brevity):

$$\underset{\theta}{\text{minimize}} \quad \frac{1}{N - T - \max(n_a, n_b) + 1} \sum_{n=\max(n_a,n_b)}^{N-T} \frac{1}{T} \sum_{k=0}^{T-1} \|y_{n+k} - \hat{y}_{n+k|n}\|_2^2,$$

$$\text{s.t.} \quad \hat{y}_{n+k|n} = h_\theta(x_{n+k|n}), \tag{3}$$

$$x_{n+k+1|n} = \text{ODEsolve}[\frac{1}{\tau} f_\theta, x_{n+k|n}, u_{n+k}, \Delta t],$$

$$x_{n|n} = \psi_\theta(u_{n-1}, ..., u_{n-n_b}, y_{n-1}, ..., y_{n-n_a}).$$

Here, the pipe ($|$) notation indicates the current index and the starting index as (current index | start index) to differentiate between different subsections. This pipe notation is similar to the notation used in Kalman filtering and conditional probability distributions (Chui et al., 2017). Furthermore, ODEsolve indicates a numerical scheme which integrates $1/\tau \, f_\theta(x, u)$ from the initial state $x_{n+k|n}$ for a length of $\Delta t$ given the input $u_{n+k}$. Lastly, we introduced an encoder function $\psi_\theta$ with encoder lengths $n_a$ and $n_b$, for the past output and input samples respectively, which estimates the initial states of the considered subsection. This encoder function will also be parameterized as a feedforward neural network during our experiments. A graphical summary of the proposed method called the CT subspace encoder approach (abbreviated as SUBNET) can be viewed in Figure 2.

The first observation is that optimization Problem (3) is a generalisation of (2) since if $T = N$ and $n_a = n_b = 0$, the original optimization Problem (2) is recovered. However, as one might observe, this optimization problem is less computationally challenging to solve if $T < N$ since the first sum can be computed in parallel. In other words, computational costs scale as $\mathcal{O}(T)$ for (3) and $\mathcal{O}(N)$ for (2). Moreover, the smoothness of the encoder cost function is also enhanced since the associated Lipschitz constant $L_{V^{(enc)}}$ can scale exponentially with the subsection length ($T\Delta t$) as shown in Theorem 1. The enhanced smoothness is reflected in the ease of optimization (Ribeiro et al., 2020).

**Theorem 1.** *The Lipschitz constant $L_{V^{(enc)}}$ of the cost function* (3)

$$\|V^{(enc)}(\theta_1) - V^{(enc)}(\theta_2)\|_2 \le L_{V^{(enc)}}\|\theta_1 - \theta_2\|_2 \tag{4}$$

*scales as*

$$L_{V^{(enc)}} = \mathcal{O}(\exp(2T\Delta t L_f/\tau)) \tag{5}$$

*where $L_f$ is the Lipschitz constant of $f_\theta$.*

*Proof.* See Appendix 8.1 □

---

[1]The adjoint methods for gradient computation is not within the scope of this research.

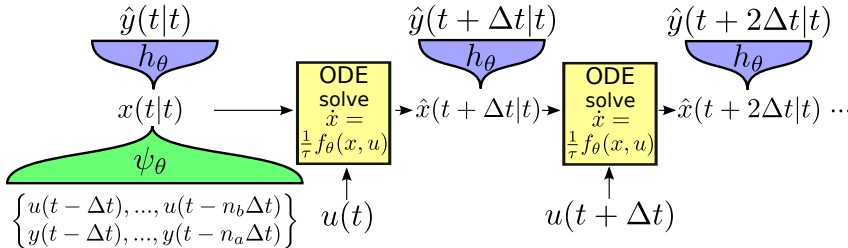

**Figure 2:** The CT subspace encoder (SUBNET) method applied on a subsection of the data of length $T\Delta t$ starting from time $t$. The encoder $\psi_\theta$ estimates the initial state, $h_\theta$ provides the output predictions while $\frac{1}{\tau}f_\theta$ governs the state dynamics. All three functions are parameterized by fully connected neural networks. Lastly, the $\frac{1}{\tau}$ factor is the novel state-derivative normalization factor which significantly increases optimization stability. All three functions are optimized together by minimizing the mean squared difference of the model outputs of multiple subsections of the available training data as seen in Eq. (3) which both reduces the computational cost and enhances cost function smoothness as seen in Theorem 1.

An error in the initial state of $x_{n|n}$ can significantly bias the estimate due to the short nature of the subsections. To counter this, we formulated an encoder function $\psi_\theta$ which estimates the initial state of each subsection. We do not add any additional loss term since an improved initial state error estimate also minimizes the transient error (Forgione & Piga, 2021a) which is present in the encoder loss.

A natural question to ask is under which conditions there exists an encoder function that can map from the past inputs and outputs to this initial state. In Appendix 8.2, we formally derive necessary conditions for the existence. These necessary conditions are that, state derivative $f_\theta$ requires to be Lipschitz continuous in $x$, and if the number of considered past outputs $n_a$ and inputs $n_b$ are equal then $n_a \geq n_x/n_y$ needs to be satisfied, among other conditions.

It is widely known that input and output normalization is essential for obtaining competitive models throughout deep learning in terms of respecting the prior assumptions made in for instance Xavier Weight Initialization (Glorot & Bengio, 2010). Input and output normalization can be seen to be insufficient when considering CT state-space model due to the presence of the hidden state $x$ and the state-derivative $f_\theta(x, u)$. However, as shown in Theorem 2, any CT system can be transformed to become normalized by the introduction of a state transform and a positive $1/\tau$ normalization factor.

**Theorem 2.** *Given $\dot{x}(t) = f(x(t), u(t))$ and $y(t) = h(x(t), u(t))$ that defines the dynamics of a system. For any bounded non-zero state-trajectory $x(t) \in \mathbb{R}^{n_x}$ and input signal $u(t) \in \mathbb{R}^{n_u}$ that satisfies $\dot{x}(t) = f(x(t), u(t))$ for all $t \in \mathbb{R}$, there exists a $\tau$ and a scalar state transformation $\gamma\tilde{x}(t) = x(t)$ such that both the equivalent state trajectory $\tilde{x}$ and state-derivative function $\tilde{f}(\tilde{x}, u)$ of the transformed system $\dot{\tilde{x}} = \frac{1}{\tau}\tilde{f}(\tilde{x}(t), u(t))$ are normalized on the time interval $[0, L]$ as*

$$RMS(\tilde{x}) = \sqrt{\frac{1}{L}\int_0^L \frac{1}{n_x}\|\tilde{x}(t)\|_2^2\, dt} = 1 \qquad \& \qquad RMS(\tilde{f}(\tilde{x}, u)) = 1. \qquad (6)$$

*if $RMS(f(x, u)) \neq 0$.*

*Proof.* With

$$\gamma = RMS(x) \qquad \& \qquad \frac{1}{\tau} = \frac{RMS(\dot{x})}{RMS(x)} \qquad (7)$$

the normalization conditions are satisfied, as shown below

$$RMS(\tilde{x}) = RMS(x)/\gamma = 1 \qquad (8a)$$

$$RMS(\tilde{f}(\tilde{x}, u)) = \tau RMS(\dot{\tilde{x}}) = \tau RMS(\dot{x})/\gamma = \tau RMS(\dot{x})/RMS(x) = 1 \qquad (8b)$$

$\square$

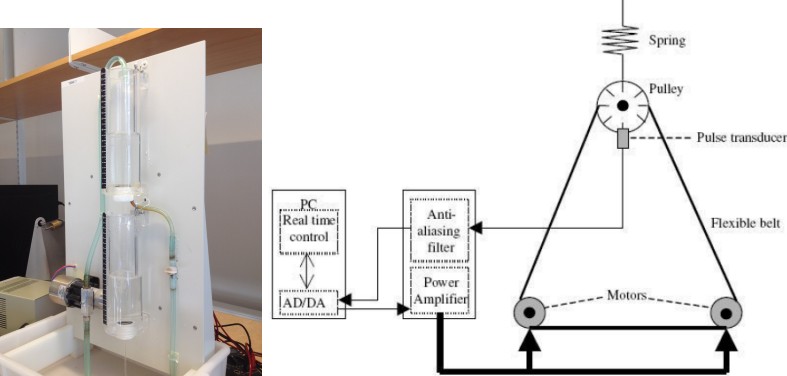

**Figure 3:** A photo of the **Cascade Tank with overflow (CCT)** system (Schoukens & Noël, 2017) and a graphical depiction of the **Coupled Electric Drive (CED)** system (Wigren & Schoukens, 2017). These two systems and the EMPS benchmark are the basis of the benchmarks used in the analysis and comparison of the SUBNET method.

Hence, to assure the existence of a properly normalized model where both the state-derivative $\tilde{f}$ function and state $\tilde{x}$ are normalized, it is sufficient to include a state and state-derivative normalization factor. Furthermore, this proof also guides the choice of $\tau$ since the amplitude of $\mathrm{RMS}(x)/\mathrm{RMS}(\dot{x})$ might be known from physical insight or by an approximate model.

## 5 EXPERIMENTS

### 5.1 BENCHMARK DESCRIPTIONS

The **Cascade Tank with overflow (CCT)** benchmark (Schoukens & Noël, 2017; Schoukens et al., 2017) consists of measurements taken from a two-tank fluid system with a pump. The input signal controls a water pump that delivers water from the reservoir to the upper tank. Through a small opening in the upper tank, the water enters the lower tank where the water level is recorded. Lastly, through a small opening in the lower tank, the water re-enters the reservoir. This benchmark is nonlinear as the flow rates are governed by square root relations and the water can overflow either tank which is a hard saturation nonlinearity. The benchmark consists of two datasets with measurements of 1024 samples each at a sample rate of $\Delta t = 4$s. The first dataset is used for training, the first 512 samples of the second set are used for validation (used only for early stopping) and the entire second set for testing. Most of the other methods to which we compare our approach use the entire second set as validation and test set, as no explicit test set is provided in this benchmark description.

The **Coupled Electric Drive (CED)** benchmark (Wigren & Schoukens, 2017) consists of measurements from a belt and pulley with two motors where both clockwise and counter-clockwise movement is permitted. The motors are actuated by the given inputs and the measured output is a pulse transducer that only measures the absolute velocity (i.e. insensitive to the sign of the velocity) of the belt. The system approximately has three states; the velocity of the belt, the position of the pulley, and the velocity of the pulley. The benchmark consists of two datasets of measured 500 samples each at a sample rate of $\Delta t = 20$ms. The first 300 samples are used for training and the other 200 samples are for testing, of those samples the first 100 samples are also used for validation with both datasets. Similar to the last benchmark, even with this overlap, it is still a fair comparison as most of the other methods to which we compare use the entire second set as validation and test.

The **Electro-Mechanical Positioning System (EMPS)** benchmark (Janot et al., 2019) consists of measured signals from a one-dimensional drive system used to drive the prismatic joint of robots or machine tools. The provided measurements of the position are obtained in closed-loop actuated and no direct velocity measurements are available. The main source of nonlinearity are the nonlinear friction effects (e.g. static and dynamic friction). The benchmark consists of two sequences of samples 24841 with a sampling time of $\Delta t = 1$ms. As prescribed by the benchmark, the first sequence is used for training and validation and the second sequence for testing (i.e. the validation set and test set are completely disjoint). Specifically for the CT subnet implementation, we utilize 17885 samples of the first set for training and the last 6956 samples are used for validation while the entire second set is used for testing.

**Table 1:** The test RMSE simulation on two benchmarks for the CT SUBNET method using an ensemble of models. The value given is the best RMSE simulation of all estimated models and the value between parentheses is the mean performance of the models. Note that we are unable to report the results for the neural ODE without state-derivative normalization $1/\tau$ for CCT since the optimization was unstable.

**(a)** CCT benchmark

| Method | RMSE |
|---|---|
| BLA (Relan et al., 2017) | 0.75 |
| Volterra model (Birpoutsoukis et al., 2018) | 0.54 |
| State-space with GP-inspired prior (Svensson & Schön, 2017) | 0.45 |
| SCI (Forgione & Piga, 2021a) | 0.40 |
| IO stable CT ANN (Weigand et al., 2021) | 0.39 |
| NL-SS + NLSS2 (Relan et al., 2017) | 0.34 |
| TSEM (Forgione & Piga, 2021a) | 0.33 |
| Tensor B-splines (Karagoz & Batselier, 2020) | 0.30 |
| Vanilla neural ODE (Chen et al., 2018) | NaN |
| neural ODE with normalization ($\Delta t/\tau = 0.03$) | 0.18 (0.33) |
| Grey-Box with physical overflow model (Rogers et al., 2017) | 0.18 |
| DT subspace encoder | 0.37 (0.97) |
| **CT subspace encoder** ($\Delta t/\tau = 0.032$) | **0.22** **(0.30)** |

**(b)** CED benchmark

| Method | RMSE [ticks/s] Set 1 | Set 2 |
|---|---|---|
| RBFNN - FSDE (Ayala et al., 2014) | 0.130 | 0.185 |
| GP with rational quadratic kernel (Zhou et al., 2021b) | 0.150 | 0.167 |
| GP with squared exponential kernel (Zhou et al., 2021b) | 0.153 | 0.132 |
| Sparse Bayesian MLP (Zhou et al., 2021b) | 0.149 (0.187) | 0.120 (0.134) |
| Cascaded Splines (Scarpiniti et al., 2015) | 0.216 | 0.110 |
| Sparse Bayesian LSTM (Zhou et al., 2021b) | 0.121 (0.155) | 0.097 (0.126) |
| Extended Fuzzy Logic (Sabahi & Akbarzadeh-T, 2015) | 0.150 | 0.092 |
| Vanilla neural ODE (Chen et al., 2018) | 0.141 (0.362) | 0.098 (0.370) |
| neural ODE with normalization ($\Delta t/\tau = 0.12$) | 0.131 (0.198) | 0.086 (0.158) |
| DT subspace encoder | 0.169 (0.187) | 0.117 (0.1557) |
| **CT subspace encoder** ($\Delta t/\tau = 0.3$) | **0.115** **(0.143)** | **0.074** **(0.100)** |

## 5.2 RESULTS

Using the SUBNET method, we estimate models where the three functions $h_\theta$, $f_\theta$ and $\psi_\theta$ are implemented as 2 hidden layer neural networks with 64 hidden nodes per layer, tanh activation and a linear bypass from the input to the output for CCT and CED and 1 hidden layer with 30 hidden nodes for EMPS. As an ODE solver, we use a single RK4 step between samples and assume that the input signal has been applied in a zero-order hold sense. As for the implementation of the CT subspace encoder-based method, the following hyperparameters are considered; $n_x = 2$, $n_a = n_b = 5$ and $T = 30$ for CCT, $n_x = 3$, $n_a = n_b = 4$ and $T = 60$ for CED and $n_x = 3$, $n_a = n_b = 20$ and $T = 200$ for EMPS. These hyperparameters are chosen based on hyperparameters analysis shown in Beintema et al. (2021) for discrete-time. We observed similar effects of the hyperparameters for continuous-time. The training is done by using the Adam optimizer with default settings (Kingma & Ba, 2015) with a batch size of 32 for CED, 64 for CCT and 1024 for EMPS and using a simulation on the validation dataset for early stopping to reduce overfitting.

We also directly compare our method with a reproduction of neural ODE on both benchmarks. We adapt the code and the example ("`latent_ODE.py`") available online (Chen et al., 2018) to include ZOH inputs, leaving the neural network unaltered and an RK4 integrator. We observed that the initial model was unstable for CCT and underperforming for CED and, hence, the neural ODE method alone was unable to provide state-of-the-art results. To stabilize and improve the neural ODE method we also introduce a state-derivative normalization term $1/\tau$ motivated by Theorem 2. The value of $1/\tau$ for CCT and CED for neural ODE was initially chosen to be the optimal value found in the SUBNET approach, however, in the CED case, it was lowered due to optimization instabilities. [2]

We compared our obtained model to the literature in Table 1 for CCT and CED. We report both the mean and the minimum of an ensemble of models estimated only differing in parameter initialization. This ensemble consists of 17 SUBNET models for both CCT and CED and 24 and 8 neuralODE models for CCT and CED respectively. The table also includes the discrete-time (DT) subspace encoder which has the same network structure and loss function as the CT subspace encoder but where the ODE solver is replaced by $f_\theta$. The table shows that the obtained models with the CT subspace encoder method provide state-of-the-art results. The obtained models are the best-known with a black-box modeling approach on both benchmarks. Furthermore, we use unrestricted

---

[2] The code used for both SUBNET and neural ODE experiments is available at `https://github.com/GerbenBeintema/CT-subnet`

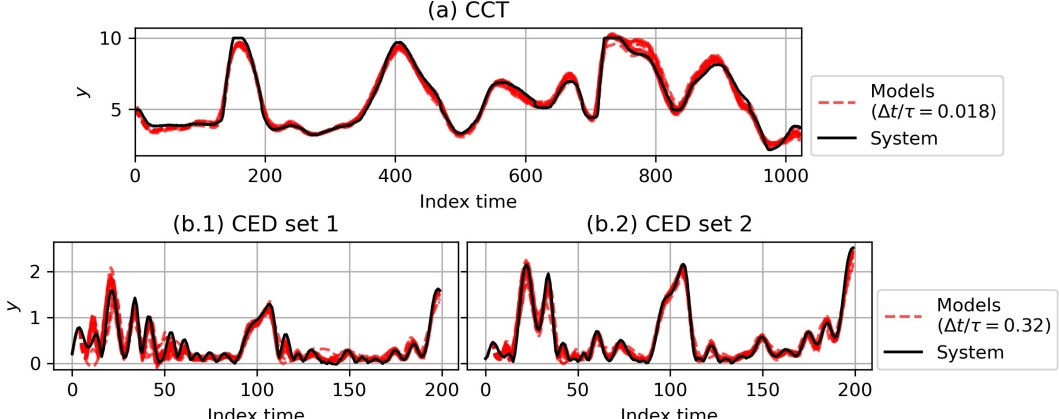

**Figure 4:** Time-domain simulation for both the (a) CCT and (b) CED benchmarks of the obtained models by the CT SUBNET method along the test set. Since the CED benchmark contains two separate test sequences, hence, they are shown in two separate figures.

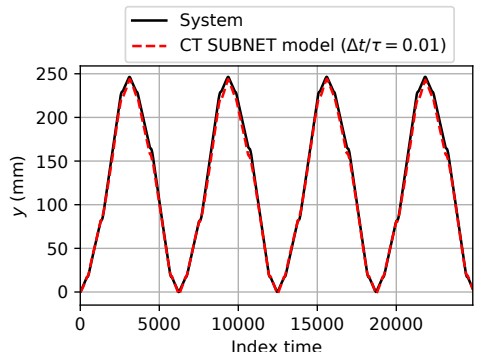

**Figure 5:** Time domain simulation results on the EMPS benchmark test set using the model obtained from the CT SUBNET method.

**Table 2:** The resulting test RMSE simulation on the test set for the EMPS benchmark compared with results in the literature.

| Method | RMSE [mm] | Fully Black-box |
|---|---|---|
| BLA (Janot et al., 2019) | 61.67 | ✓ |
| dynoNET (Forgione & Piga, 2021b) | 2.64 | ✗ |
| IO stable CT ANN (Weigand et al., 2021) | 3.68 | ✓ |
| TSEM (Forgione & Piga, 2021a)[a] | 4.73 | ✗ |
| SCI (Forgione & Piga, 2021a)[a] | 6.65 | ✗ |
| Vanilla neural ODE (Chen et al., 2018) | NaN | ✓ |
| **CT subspace encoder** ($\Delta t/\tau = 0.01$) | 4.61 | ✓ |

[a]RMS values reported in Weigand et al. (2021)

state-space and fully connected neural networks as model elements. Remarkably, the resulting performance is close to the performance of a grey-box model. Furthermore, Figure 4 illustrates that the resulting models have been able to model the nonlinear behavior present in both benchmarks.

Table 1 also contains the results of the modified neural ODE with normalization. One observation is that the best and mean performance difference is significantly larger than for SUBNET. We think that this is due to the availability of only a single sequence in the training set for the CCT benchmark (and two sequences for CED) which results in extensive overfitting. In comparison, the subspace encoder method is less prone to overfitting since it uses many subsections of the available sequence(s). Moreover, the subspace encoder method only requires about 20 minutes to train a model to the lowest validation loss, whereas, neural ODE requires about 2 hours for CED and 5 hours for CCT.

We compare the CT SUBNET method applied on the EMPS benchmark to the existing methods in Table 2 and show the simulated response on the test set in Figure 5. The obtained results show a remarkable accuracy over the entire $24841$ samples showing that the CT SUBNET method is able to make accurate long-term predictions while using relatively short sub-sequences of only $T = 200$ in length during training. In the table, dynoNET is significantly better than the proposed method, however, this method utilizes grey-box knowledge (i.e. physics-based) in the network structure but this is system specific and not easily generalizable. Lastly, IO stable CT ANN performs slighly better than the proposed method since it enforces stability which can be a problem during estimation since there is a position integrator present in system. Furthermore, since the validation and test set are disjoint we also show that overfitting does not play a role in the reported results.

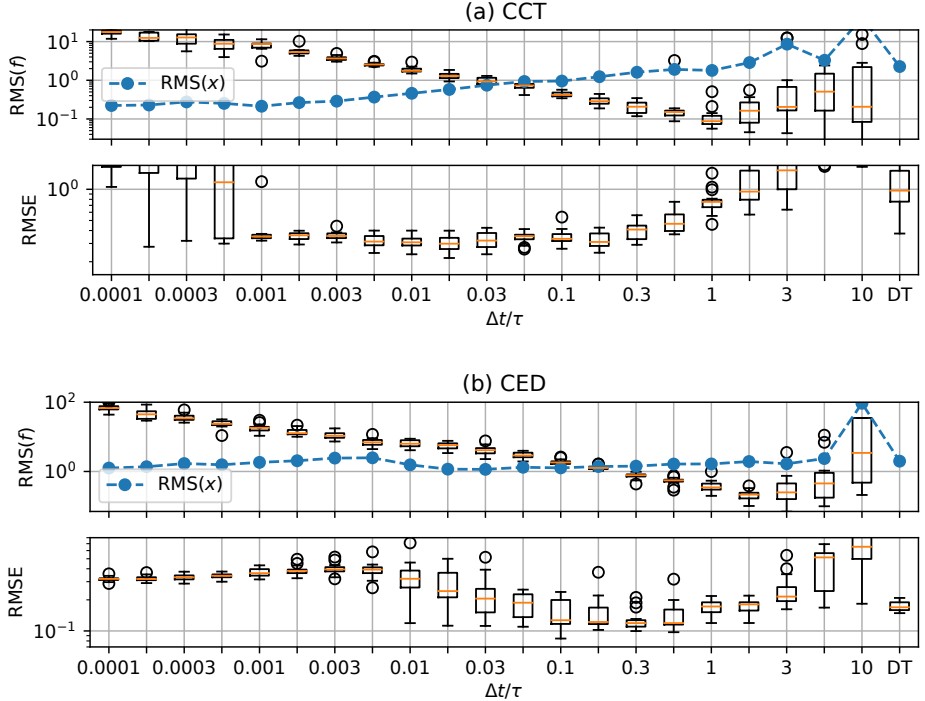

**Figure 6:** The influence of the state-derivative normalization hyperparameter $\Delta t/\tau$ as in $\dot{x} = \frac{1}{\tau} f_\theta(x, u)$ on different model properties for both the (a) CCT and (b) CED benchmarks. The shown model properties are the mean state amplitude $\text{RMS}(x)$, mean state-derivative amplitude $\text{RMS}(f)$ and the RMSE simulation on the test set(s). These two figures show that there exists a range of $\Delta t/\tau$ where $\text{RMS}(f) \approx \text{RMS}(x) \approx 1$ which numerically validates Theorem 2. Furthermore, $\Delta t/\tau$ with this property also has a significantly lowered RMSE simulation as was argued in its introduction.

When we introduced the state-derivative normalization factor $1/\tau$, we argued that it would normalize $f_\theta$ and that it would increase optimization stability and, hence, the quality of the obtained models. Here, we provide some empirical insight for these two statements by providing a parameter sweep over $\Delta t/\tau$. To eliminate variations due to different initial parameters we trained an ensemble of models which creates box plots with mean state amplitude defined by $\text{RMS}(x) \triangleq \sqrt{\frac{1}{N n_x} \sum_k \|x(k\Delta t)\|_2^2}$, mean state-derivative amplitude $\text{RMS}(f)$, and the RMSE simulation. These box-plots as shown in Figure 6 indeed illustrate that there exists an $1/\tau$ such that both $\text{RMS}(f) \approx \text{RMS}(x) \approx 1$ and that the best performing models are close to that value of $1/\tau$. Moreover, to illustrate that improper normalization (i.e. $\tau = 1$) can diminish the performance for both CCT and CED, observe that the RMSE simulation on the test set(s) is 2.0 and 0.3 [ticks/s] for $\Delta t/\tau = \Delta t = 4$ s and $\Delta t/\tau = \Delta t = 0.02$ s respectively.

## 6 CONCLUSION

In this paper, we have introduced the CT subspace encoder approach to identify nonlinear dynamical systems in the presence of latent states, external inputs, and measurement noise. We have shown that the proposed method can obtain highly accurate CT models only consisting of fully connected neural networks. The approach has improved computational cost and stability by considering multiple subsections where the initial state is estimated with an encoder function and by a state-derivative normalization term to improve optimization stability. We provided multiple theoretical proofs which provide additional insight and motivation for the method. These proofs are, increased cost function smoothness, necessary conditions for the existence of the encoder function and that for proper normalization in CT modeling one requires the state-derivative normalization term. Furthermore, we obtain state-of-the-art results on all three considered benchmarks.

## 7 REPRODUCIBILITY STATEMENT

- **Datasets:** *(i)* The CCT dataset is described in Schoukens & Noël (2017); Schoukens et al. (2017) and is available for download at `https://data.4tu.nl/articles/dataset/Cascaded_Tanks_Benchmark_Combining_Soft_and_Hard_Nonlinearities/12960104`, *(ii)* the CED dataset is described in Wigren & Schoukens (2017) and is available for download at `http://www.it.uu.se/research/publications/reports/2017-024/`, *(iii)* the EMPS datset is described in (Janot et al., 2019) and is available for download at `https://www.nonlinearbenchmark.org/benchmarks/emps`.

- **Code:** Both the implementation and experiments of CT SUBNET and neural ODE are available at `https://github.com/GerbenBeintema/CT-subnet`.

- **Hardware:** It takes about 15 minutes to estimate a single CT SUBNET model and 2 hours for a single neural ODE model on a consumer laptop. A notable exception is CT SUBNET for EMPS which took about 10 hours due to the increased size and difficulty of the dataset.

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

## 8 APPENDIX

### 8.1 PROOF OF THEOREM 1

Recall that the subspace encoder loss function as in Eq. 3 can be expressed in the following form

$$
\begin{aligned}
V^{\text{enc}}(\theta) = \quad & \frac{1}{N - T - \max(n_a, n_b) + 1} \sum_{n=\max(n_a,n_b)}^{N-T} \frac{1}{T} \sum_{k=0}^{T-1} \|y_{n+k} - \hat{y}_{n+k|n}\|_2^2, \\
\text{with} \quad & \hat{y}_{n+k|n} = h_\theta(x_{n+k|n}) = h_\theta(x((n+k)\Delta t | n\Delta t)), \\
& \dot{x}(t|n\Delta t) = \frac{1}{\tau} f_\theta(x(t|n\Delta t), u(t)) \\
& x(n\Delta t | n\Delta t) = \psi_\theta(u_{n-1}, ..., u_{n-n_b}, y_{n-1}, ..., y_{n-n_a}).
\end{aligned}
\tag{9}
$$

Our aim is to derive the scaling in $T$ of the Lipschitz constant $L_{\text{enc}}$ as defined in

$$
|V^{\text{enc}}(\theta_1) - V^{\text{enc}}(\theta_2)|^2 \le L_{\text{enc}}^2 \|\theta_1 - \theta_2\|_2^2, \quad \forall \theta_1, \theta_2 \in \Theta \subset \mathbb{R}^{n_\theta}.
\tag{10}
$$

We aim to express $L_{\text{enc}}$ in terms of the following Lipschitz constants

$$
\|h_{\theta_1}(x_1) - h_{\theta_2}(x_2)\|_2^2 \le L_h^2 (\|x_1 - x_2\|_2^2 + \|\theta_1 - \theta_2\|_2^2).
\tag{11a}
$$

$$
\|f_{\theta_1}(x_1, u) - f_{\theta_2}(x_2, u)\|_2^2 \le L_f^2 (\|x_1 - x_2\|_2^2 + \|\theta_1 - \theta_2\|_2^2),
\tag{11b}
$$

$$
\|\psi_{\theta_1}(u_{\text{past}}, y_{\text{past}}) - \psi_{\theta_2}(u_{\text{past}}, y_{\text{past}})\|_2^2 \le L_\psi^2 \|\theta_1 - \theta_2\|_2^2.
\tag{11c}
$$

for all $x_1, x_2 \in X \subset \mathbb{R}^{n_x}$ and $\forall \theta_1, \theta_2 \in \Theta \subset \mathbb{R}^{n_\theta}$. A sufficient condition for these Lipschitz constants to be finite is that the derivatives are finite on a compact set of inputs which is often the case in feed-forward neural networks.

Furthermore, to derive the scaling of $L_{\text{enc}}$ we use known properties of the Lipschitz constants;

- The sum property: $c(x) = a(x) + b(x)$ has a Lipschitz constant of $L_c = L_a + L_b$.
- The multiplication property: $c(x) = a(x)b(x)$ has a Lipschitz $L_c = M_a L_b + M_b L_a$ where $M_a$ is the maximal value of $a$ on a closed set of inputs $x \in X$ and $M_b$ being similarly defined.

Since the encoder loss function as in Eq. 3 can be written as a sum:

$$V^{\text{enc}}(\theta) = \frac{1}{N - T - \max(n_a, n_b) + 1} \sum_{n=\max(n_a,n_b)}^{N-T} V^{\text{sec}}(n, \theta_1) \tag{12a}$$

$$V^{\text{sec}}(n, \theta_1) = \frac{1}{T} \sum_{k=0}^{T-1} \|y_{n+k} - \hat{y}_{n+k|n}\|_2^2 \tag{12b}$$

where

$$|V^{\text{sec}}(n, \theta_1) - V^{\text{sec}}(n, \theta_2)|^2 \leq L_{\text{sec}}^2 \|\theta_1 - \theta_2\|_2^2, \tag{12c}$$

Then by the sum property this implies that $L_{\text{enc}} = L_{\text{sec}}$ since

$$L_{\text{enc}} = \frac{1}{N - T - \max(n_a, n_b) + 1} \sum_{n=\max(n_a,n_b)}^{N-T} L_{\text{sec}}. \tag{13}$$

Hence, it is sufficient to consider only a single subsection. Take $n = 0$ and drop the bar notation for simplicity. By the sum and multiplication properties, we derive get

$$|V^{\text{sec}}(\theta_1) - V^{\text{sec}}(\theta_2)| \leq 2/T \sum_{k=0}^{T-1} (M_y + M_k)\|\hat{y}_{1,k} - \hat{y}_{2,k}\|_2, \tag{14}$$

where $M_y$ is the bound on $\|y(t)\|_2$ assuming a stable system and $M_k$, the bound on $\|\hat{y}_k\|_2$. The $M_k$ bound scales the same as $\|\hat{y}_{1,k} - \hat{y}_{2,k}\|_2$ as shown in Ribeiro et al. (2020). The $\|\hat{y}_{1,k} - \hat{y}_{2,k}\|_2$ expression can be expanded by using Eq. 11a as

$$\|\hat{y}_{1,k} - \hat{y}_{2,k}\|_2^2 \leq L_h^2 (\|x_1(k\Delta t) - x_2(k\Delta t)\|_2^2 + \|\theta_1 - \theta_2\|_2^2). \tag{15}$$

Next, we aim to derive an expression for the Lipschitz constant $L_x(t)$ given in terms of

$$\|x_1(t) - x_2(t)\|_2^2 \leq L_x(t)^2 \|\theta_1 - \theta_2\|_2^2. \tag{16}$$

whereby Eq. (11c)

$$L_x(0) = L_\psi. \tag{17}$$

By considering a small increment in time of length $h$, we can express $L_x(t + h)$ in terms of $L_x(t)$. Using the fact that $h$ is small we can use an Eurler step and discard higher order terms of $h$ as;

$$\|x_1(t+h) - x_2(t+h)\|_2^2 = \|(x_1(t) - x_2(t)) + h/\tau(f_{\theta_1}(x_1(t), u(t)) - f_{\theta_2}(x_2(t), u(t)))\|_2^2$$

$$\leq \|x_1(t) - x_2(t)\|_2^2 + 2h/\tau\|x_1(t) - x_2(t)\|_2 \|f_{\theta_1}(x_1(t), u(t)) - f_{\theta_2}(x_2(t), u(t))\|_2$$

by the triangle inequality. Next, we can replace all the $f$ terms by Eq. (11b) and $x$ terms by Eq. (16) to derive an expression for $L_x(t + h)$ as

$$\leq \|x_1(t) - x_2(t)\|_2^2 + 2h/\tau\|x_1(t) - x_2(t)\|_2 L_f \sqrt{\|x_1(t) - x_2(t)\|_2^2 + \|\theta_1 - \theta_2\|_2^2}$$

$$\leq \left( L_x(t)^2 + 2h/\tau L_x(t) L_f \sqrt{L_x(t)^2 + 1} \right) \|\theta_1(t) - \theta_2(t)\|_2^2$$

$$L_x(t+h) = \sqrt{L_x(t)^2 + 2h L_x(t)\sqrt{L_x(t)^2 + 1} L_f/\tau}.$$

This expression allows us to derive that the derivative of $L_x(t)$ is given by

$$\dot{L}_x(t) = \lim_{h \to 0} \frac{L_x(t+h) - L_x(t)}{h}, \tag{18}$$

$$= \sqrt{1 + L_x(t)^2} L_f/\tau. \tag{19}$$

which suggests that $\dot{L}_x(t)$ is continuous in $t$ and has the closed form solution as

$$L_x(t) = L_x(0) + \int_0^t \dot{L}_x(t')dt' \tag{20}$$

$$= \sinh(tL_f/\tau + \operatorname{arcsinh}(L_\psi)) \tag{21}$$

Now by substituting Eq. (21) into, (16), (15), (14) and using (13) we arrive at the following expression for $L_{\text{enc}}$ as

$$L_{\text{enc}} = 2/T \sum_k (M_y + M_k)L_h(\sinh(k\Delta t L_f/\tau + \operatorname{arcsinh}(L_\psi)) + 1) \tag{22a}$$

which scales in the limit of large $T$ as

$$L_{\text{enc}} = \mathcal{O}(\exp(2T\Delta t L_f/\tau)), \tag{23a}$$

since $L_f > 0$ and $\Delta t > 0$. Note that the 2 in the exponent is from the multiplication with $M_k$ which also scales as the $y$ term as previously mentioned.

Furthermore, this bound cannot be lowered since the linear system $\dot{x}(t) = x(t)L_f/\tau$ already results in the scaling of $L_{\text{enc}} \sim \exp(2T\Delta t L_f/\tau)$.

## 8.2 RECONSTRUCTABILITY OF THE INITIAL STATE FROM PAST INPUT AND OUTPUTS

To derive the conditions on the existence of the encoder function suppose that we have a system given by

$$\dot{x}(t) = f(x(t), u(t)), \tag{24a}$$

$$y_n = h(x_n) + w_n, \tag{24b}$$

where $x_n = x(n\Delta t)$. For this system, if a state is given $x(t_0)$ along the input trajectory $u(t)$ one would in principle be able to compute $x(t)$ for all $t > t_0$. However, since we aim to construct the state given past outputs we need $x(t)$ for $t < t_0$ which requires backward in time integration. This backward integration on $f$ is guaranteed to be unique if $f$ is Lipschitz continuous in $x$ for all $u$ as by the Picard–Lindelöf theorem (Murray & Miller, 2013). Hence, since $f$ is assumed to be Lipschitz continuous we can construct an operator $f_d$ which can integrate backwards or forwards as

$$x_{n+1} = f_d(x_n, u_n) \rightarrow x_{n-1} = f_d^{-1}(x_n, u_{n-1}) \tag{25}$$

where $u$ is subject to ZOH.

This operator allows us to construct past outputs as

$$
\begin{aligned}
y_{n-1} &= (h \circ f_d^{-1})(x_n, u_{n-1}) + w_{n-1} \\
y_{n-2} &= (h \circ f_d^{-2})(x_n, u_{n-1}, u_{n-2}) + w_{n-2} \\
&\vdots \\
y_{n-z} &= (h \circ f_d^{-z})(x_n, u_{n-1}, u_{n-2}, ..., u_{n-z}) + w_{n-z} \\
Y_n^{-z} &= (H \circ F_d^{-z})(x_n, U_n^{-z}) + W_n^{-z}
\end{aligned}
\tag{26}
$$

where

$$f_d^{-p}(x_n, u_{n-1}, ..., u_{n-p}) = f_d^{-p+1}(f_d^{-1}(x_n, u_{n-1}), u_{n-2}, ..., u_{n-p})$$

is the application of $f_d^{-1}$, $p$ times to obtain $x_{n-p}$. Furthermore, $Y_n^{-z} = [y_{n-1}^\top, y_{n-2}^\top, ..., y_{n-z}^\top]^\top$ and, $(H \circ F_d^{-z})$ with $W_n^{-z}$ are similarly defined. To construct the initial state $x_n$, we need to invert Eq. (26). This inverse is also known as a reconstructability map (Katayama, 2005). For the inverse to exist, several necessary requirements can be given. One such necessary requirement is that a small perturbation to a solution $x_n$ should change $(H \circ F_d^{-z})$, otherwise these solutions are indistinguishable from the output. This is formalized stating by that the matrix $\frac{\partial (H \circ F_d^{-z})(x_n, U_n^{-z})}{\partial x_n}$ has a null space of rank zero which is also known as the local observability condition. This is the same as the condition that the column rank of this matrix $\geq n_x$. A necessary requirement for this

column rank condition is that the number of columns is equal to or greater than the number of rows i.e. $z n_y \geq n_x$.

Hence, under the right conditions, it might be possible to solve Eq. (26) for a singular $x_n$ since this equation is a nonlinear fixed point problem if $W_n^{-z}$ is known. For the case that $W_n^{-z}$ is unknown one can estimate the state $\hat{x}_n \approx x_n$ by solving the nonlinear regression problem;

$$\hat{x}_n = \arg\min_{\hat{x}_n} \|Y_n^{-z} - (H \circ F_d^{-z})(\hat{x}_n, U_n^{-z})\|_2, \tag{27a}$$

$$= \arg\min_{\hat{x}_n} \|L(\hat{x}_n, Y_n^{-z}, U_n^{-z})\|_2. \tag{27b}$$

Hence, both $f$ uniformly Lipschitz continuous and $(\nabla_x L)^T \nabla_x L$ being full rank in $\hat{x}_n$ are necessary conditions for the existence of a unique reconstructability map.

Computing the reconstructability map for our model thus requires solving an optimization problem that becomes computationally infeasible during training. Hence, the encoder function aims to approximate the solution to Problem (27).

