# OpenReview forum: "Continuous-time identification of dynamic state-space models by deep subspace encoding"
_ICLR.cc/2023/Conference — ICLR 2023 poster_

### Official Review · Reviewer_jYDD · 2022-10-24

**Confidence:** 2
**Correctness:** 3
**Technical Novelty And Significance:** 2
**Empirical Novelty And Significance:** 2
**Recommendation:** 5

**Clarity, Quality, Novelty And Reproducibility:**


The paper lacks a bit of clarity, especially the description of the methodology and the training of the neural networks. The results of the proposed approach seem identical to the neural ODE with normalization, but at a much lower computational cost.
The authors intend to open source the code for reproducibility purpose.


**Strength And Weaknesses:**

Strengths:
1.	The authors tested their approach on two different datasets.
2.	The authors have developed proof of convergence of their technique.
3.	The authors have provided source code for better reproducibility purpose.
Weaknesses:

1.	The results in table 1 seems to suggest that for both problems the neural ODE approach with normalization seems to be performing as well or even better than the proposed technique. Do the authors suggest that the main advantage of their approach lies in the lower estimation time (lower computation complexity).
2.	I fail to understand how the proposed approach works, especially the joint training of 3 fully-connected neural networks.
3.	Figure 5 caption could be expanded for better understanding.

Additional questions:
Have the authors analyzed the effect of the subsections size on the performance of the technique?
Why did the authors compare their technique with different approaches on the two datasets (for example BLA (Relan et al. 2017) and Volterra model only assessed on the CCT benchmark)?
Could the authors comment on how the proposed technique compare to the Neural Laplace (“Neural Laplace: Learning diverse classes of differential equations in the Laplace domain”, Samuel Holt et al. ICML 2022) ?



**Summary Of The Paper:**

This paper introduces a novel approach for the estimation of continuous time state space dynamical models.
This field has gained significant attraction since 2018 with the Neural ODE.
The proposed approach is based on the evaluations on short subsections, and a novel state-derivative normalization.

**Summary Of The Review:**

The paper introduces a novel approach for the identification of continuous dynamical state-space model parameters. The method is interesting, but I fail to understand the training process. The results seem to indicate performance on par with “state-of-the-art” approach but a lower computational cost.

---

> ### Author Response · Authors · 2022-11-14
> **Author Response to Reviewer jYDD**
>
> Thank you very much for carefully reading our paper and for your detailed comments, concerns and feedback.
>
> **"The results in table 1 seems to suggest that for both problems the neural ODE approach with normalization seems to be performing as well or even better than the proposed technique. Do the authors suggest that the main advantage of their approach lies in the lower estimation time (lower computation complexity)."** Indeed, neural ODE with normalization is on par with the CT subspace while looking at the best RMSE simulation. However, as mentioned in the text, the mean RMSE simulation of the ensemble of models (the number between parentheses) is higher in both cases. This suggests, as mentioned in the text and seen during training, that the neural ODE approach is less consistent in obtaining great results compared to the CT SUBNET. This is, as we argue, related to enhanced cost function smoothness as shown in Theorem 1. Furthermore, without the added normalization $1/\tau$ which is one of the contributions of this work, the neural ODE does not result in good estimates at all.
>
> **"I fail to understand how the proposed approach works, especially the joint training of 3 fully-connected neural networks."** Thanks for the comment, we indeed recognize now that there is quite a leap in the thought present in the draft on why the minimization of Eq. (3) would optimize all three networks to provide a "good" model of the system. Intuitively, since Eq. (3) is a generalization of the problem (2) this suggests that it can train $f\_\theta$ and $h\_\theta$. Furthermore, as mentioned in the text, we do not need to add any additional loss term for the encoder since an improved initial state error estimate also minimizes the transient error which is present in eq. (3) and hence the overall cost.
>
> **"Figure 5 caption could be expanded for better understanding."** Have re-written the caption of figure 6 (changed from 5 to 6 due to added figure in between) as: "*The influence of the state-derivative normalization hyperparameter $\Delta t/\tau$ as in  $\dot{x} = \frac{1}{\tau} f\_\theta(x,u)$ on different model properties for both the (a) CCT and (b) CED benchmarks. The shown model properties are the mean state amplitude $\text{RMS}(x)$, mean state-derivative amplitude $\text{RMS}(f)$ and the RMSE simulation on the test set(s). These two figures show that there exists a range of $\Delta t/\tau$ where $\text{RMS}(f) \approx \text{RMS}(x) \approx 1$ which
> numerically validates Theorem 2. Furthermore, $\Delta t/\tau$ with this property also has a significantly lowered RMSE simulation as was argued in it introduction of this hyperparameter.*"
>
> **"Have the authors analyzed the effect of the subsections size on the performance of the technique?"** Interesting question, the subsection's size can significantly alter the performance of the method which was shown by (Beintema et. al., 2021) in their analysis in discrete time. We observe similar effects on performance for our case of CT SUBNET. Thus we do not see it necessary to include an in-depth analysis and rather refer the reader to the analysis performed in Beintema et. al. 2021. Submission change: _"These hyperparameters are chosen based on hyperparameters analysis shown in Beintema et. al., 2021 for discrete-time. We observed similar effects of the hyperparameters for continuous-time."_
>
> **"Why did the authors compare their technique with different approaches on the two datasets (for example BLA (Relan et al. 2017) and Volterra model only assessed on the CCT benchmark)?"** Since these two benchmarks are publicly available, most of the RMS values reported in Table 1 were obtained using the values reported in other publications and most without publicly available code. Hence, not all publications considered both benchmarks and, thus the discrepancies between the two tables. However, due to the large diversity of approaches present a broad comparison is still ensured with respect to both classical and highly novel approaches for the data-driven modelling of dynamical (engineering) systems.
>
> **"Could the authors comment on how the proposed technique compare to the Neural Laplace (“Neural Laplace: Learning diverse classes of differential equations in the Laplace domain”, Samuel Holt et al. ICML 2022)?"** I was unaware of this work but it is really interesting. Their approach can also model DDE which is not considered in this work. The largest difference is that our method directly aims to estimate the state-space equations and output function whereas Neural Laplace aims to find the Laplace representation. Furthermore, a direct comparison might be quite involved since, as far as I could see from the paper, there are no hidden states or noise present in the provided numerical examples. However, these issues are not insurmountable and a comparison is warranted to be made in future research.

---

### Official Review · Reviewer_mnoy · 2022-11-01

**Confidence:** 2
**Correctness:** 4
**Technical Novelty And Significance:** 3
**Empirical Novelty And Significance:** Not applicable
**Recommendation:** 8

**Clarity, Quality, Novelty And Reproducibility:**

Clarity: Beyond the comments above, the methodology and experiment description of the manuscript is clear to read and easy to follow.

Quality: The quality of the manuscript is good, with sufficient algorithm analysis and insights. Experiment results, especially the comparison with state-of-art methods, also validate the author’s claims.

Reproducibility: Based on the reproducibility statement provided by the manuscript, the proposed model and the experiment shall be reproducible.


**Strength And Weaknesses:**

Strength: The proposed work provides a solid foundation to improve the system identification modeling by the derivation of the necessary condition for encoding the initial state when performing system identification on overlapping short subsections and proving that the smoothness of the encoder cost function will be improved by doing so. Model performance shown in the experiment results, especially the comparison with neural ODE, further validates the effectiveness of the proposed model.

Weakness: The structure of the SUBNET is unclear from Figure 2 and the corresponding text. As SUBNET is the key contribution of this work, more details are welcomed. The reviewer is also curious about how the discrete-time subspace encoder, as listed in Table 1, is implemented.


**Summary Of The Paper:**

The paper proposed a new scheme for the continuous-time nonlinear state-space system identification based on overlapping short subsections rather than the whole length of time, with the help of a subspace encoder and state-derivative normalization. The paper proved that, both algorithmically and empirically, the proposed scheme will reduce computational complexity and improve cost function smoothness.

**Summary Of The Review:**

A solid and novel scheme for the continuous-time nonlinear state-space system identification.

---

> ### Author Response · Authors · 2022-11-14
> **Author Response to Reviewer mnoy**
>
> Thank you for your comments and feedback.
>
> **"The structure of the SUBNET is unclear from Figure 2 and the corresponding text. As SUBNET is the key contribution of this work, more details are welcomed."** To clarify better the structure of the proposed network we added additional information to the caption of the figure on the cost function and the benefits of the proposed method. The new caption reads: "*The CT subspace encoder (SUBNET) method applied on a subsection of the data of length $T\Delta t$ starting from time $t$. The encoder $\psi\_\theta$ estimates the initial state, $h\_\theta$ provides the output predictions while $\frac{1}{\tau} f\_\theta$ governs the state dynamics. All three functions are parameterized by fully connected neural networks. Lastly, the $\frac{1}{\tau}$ factor is the novel state-derivative normalization factor which significantly increases optimization stability. All three functions are optimized together by minimizing the mean squared difference of the model outputs of multiple subsections of the available training data as seen in Eq. (3) which both reduces the computational cost and enhances cost function smoothness as seen in Theorem 1.*"
>
> **"The reviewer is also curious about how the discrete-time subspace encoder, as listed in Table 1, is implemented."** To clarify the discrete-time implementation, additional explanations have been provided in the paper. In the implementation, the discrete-time subspace encoder has the same network structure and loss function as the CT subspace encoder but $f\_\theta$ simplifies to the state-transition function as $x(t+\Delta t) = f\_\theta(x(t), u(t))$.

---

### Official Review · Reviewer_x1Z3 · 2022-11-04

**Confidence:** 3
**Correctness:** 4
**Technical Novelty And Significance:** 2
**Empirical Novelty And Significance:** 3
**Recommendation:** 5

**Clarity, Quality, Novelty And Reproducibility:**

**Novelty**: As the authors claim, the novel aspects of the framework includes:

1. generating initial conditions for each sequence section using an encoder function as another neural network independently optimized using past observations;
2. introducing a normalization term tau for better performance;

**Clarity & Quality:**

Unfortunately, the benefits as a result of both novel aspects are not clearly explained.

1. Specifically, while it is reasonable to expect that having such an encoder function for separately optimizing initial hidden states can increase stability of the full optimization, it is not clear at all how this invention can help to reduce overall computational complexity.
2. Supposedly, the normalization term tau plays a role of regularization for better training stability, but how it helps is not clear.

**Strength And Weaknesses:**

**Strength:**

Having the original joint optimization problem broken down to 1) optimizing initial hidden state and 2) optimizing forward dynamics have the potential to increase optimization stability.

**Weaknesses:**

Since the framework uses three neural networks as function approximators for encoder, iterative dynamics, and output projection, the full model and optimization scheme look very similar to training a standard recurrent neural network. From this comparison, it is less clear about the benefits of having such elaborate framework in terms of increasing performance and reducing computational complexity.

**Summary Of The Paper:**

1. The authors try to infer the underlying dynamics of some unknown dynamical system based on sparsely sampled observations and external inputs.
2. The authors build a framework to infer such a dynamical system using 1) neural networks as function approximators for a generic continuous-time dynamical system, and 2) gradient-based optimization.
3. The framework can account for a dynamical system that demands 1) handling noisy measurements, 2) doing long term prediction, 3) compatible with hidden states, and 4) compatible with external inputs.
4. The quality of the inferred model is evaluated by how good the observed trajectories are reconstructed in two benchmark problems.

**Summary Of The Review:**

1. I will recommend this paper if the author address and clarify the above issues.
2. Without such clarification, it is hard to judge whether this work add to the existing effective approach: e.g., training RNN with backprop through time, or doing FORCE learning with chaotic or balanced neural network.

---

> ### Author Response · Authors · 2022-11-16
> **Author Response to Reviewer x1Z3**
>
> Thank you very much for reading our paper and for your comments and concerns.
>
> **"Since the framework uses three neural networks as function approximators for encoder, iterative dynamics, and output projection, the full model and optimization scheme look very similar to training a standard recurrent neural network. From this comparison, it is less clear about the benefits of having such elaborate framework in terms of increasing performance and reducing computational complexity."**
>
> * The main difference with recurrent neural network is that we consider the problem of estimating a continuous-time state-space model as in $\frac{d x(t)}{dt} = f\_\theta(x(t),u(t))$, $\hat{y}(t) = h\_\theta(x(t))$ Eq. (2) which has multiple challenges if directly trained using conventual methods (e.g. see neuralODE paper and Eq. (2)). i.e.:
>   * The computational cost of computing the loss scales with data sequence length $O(N)$
>   * The cost function can become exponentially non-smooth for increasing length of simulation (See theorem 1) which can makes gradient-based optimization unreliable.
> * Thus we investigate how one could use many small subsections of length $T$ of the available data during training without compromising on long-term prediction capabilities.
>   * This reduces the computational cost to $O(T)$ since it allows for parallelization over multiple subsections.
>   * And increases cost function smoothness  (See theorem 1) and thus makes gradient-based optimization more reliable.
> * In the paper we recognise that incorrect initial states at the start of these subsections can significantly interfere with model estimation, hindering the long-term prediction capabilities of the estimated model. To negate this effect we introduce a subspace encoder function $\psi_\theta$ which aims to approximate the initial state. This function is trained together with $f\_\theta$ and $h\_\theta$ and does not require any additional loss terms to be added. Furthermore, we provide some additional insight by providing conditions for the existence of this encoder function (see Appendix 8.2).
> * The proposed model is applied on three benchmarks which show state-of-the-art results in a reproducible and reliable manner.
>   * CCT benchmark: 0.22 CT SUBNET vs NaN neural ODE (did not converge)
>   * CED benchmark: 0.115 & 0.074 CT SUBNET vs 0.141 & 0.098 neural ODE
>   * EMPS benchmark: 4.61 CT SUBNET vs NaN neural ODE (did not converge)
> * Remarkably, the EMPS benchmark result (Figure 5) shows that the CT SUBNET is able to obtain long-term prediction capabilities since it could get accurate prediction over the entire 24841 horizon while only trained on sequences of length 200.
>
> In summary, the proposed method reduces the computational cost from $O(N)$ to $O(T)$ (length of a subsection) while increasing optimization stability and reliability while giving accurate long-term predictions on all three considered benchmarks.
>
> With your input, we have adjusted the paper in a number of places to improve the clarity of the paper, e.g. we added or altered: "_In other words, computational costs scale as $O(T)$ for Eq. (3) and $O(N)$ for Eq. (2)._", we made adjustments in the caption of figure 2, clarified the role of the encoder, made the novelty of the method more clear, corrected a number of small typos, among other improvements.
>
> **"Supposedly, the normalization term tau plays a role of regularization for better training stability, but how it helps is not clear."**
>
> It is well-recognized that the normalization of inputs and outputs when training feed-forward neural networks is often essential. The model structure includes both a hidden state $x(t)$ and ANN output $f(x(t),u(t))$. We show with Theorem 2 that $x(t)$ and $f(x(t),u(t))$ cannot both be normalized (i.e. average amplitude of 1) at the same time for most systems considering only a scalar state transformation. Thus, we introduce an additional normalization constant $\tau$ which allows for both $x(t)$ and $f(x(t),u(t))$ to be normalized at the same time. See theorem 2 for details on how this is possible.
>
> In our benchmark results, we show that the introduction of $\tau$ is essential to obtain good results for continuous-time neural network approaches. It also drastically improved the results obtained for the well-established ODENet approach. Furthermore, we numerically validate our theorem 2 in figure 6 which indeed shows that there exists a $\tau$ such that both $x(t)$ and $f(x(t),u(t))$ are normalized at the same time.
>
> We added additional clarification to the introduction of the $\tau$ normalization constant. e.g. in "*It is widely known that input and output normalization is essential for obtaining competitive models throughout deep learning such to respect to the prior assumptions made in for instance Xavier Weight Initialization (Glorot & Bengio, 2010).*" and completely rewrote the caption of figure 6 (originally it was version figure 5) to clarify the role $\tau$ from a practical standpoint.

---

### Official Review · Reviewer_mcyj · 2022-11-05

**Confidence:** 3
**Correctness:** 3
**Technical Novelty And Significance:** 3
**Empirical Novelty And Significance:** 3
**Recommendation:** 6

**Clarity, Quality, Novelty And Reproducibility:**

The work is very clearly written and the quality of the writing is high.   As noted above, the idea of breaking up longer sequences into shorter ones and inferring initial state for each shorter sequences is, to my knowledge, novel and an important contribution.  I have no concerns about the reproducibility of the results.

**Strength And Weaknesses:**

The paper is well written and, as the authors do a very good job of motivating, the problem of fitting continuous models to the behavior of dynamical systems is important.  The authors also do a good job of contextualizing their work with regards to the existing literature.   While the idea of using Euler integration followed by back propagation to learn neural network descriptions of differential equations is not novel,
I believe the idea of approaching this through (1) breaking up longer sequences into shorter ones and (2) using a neural network to predict initial state at the beginning of each shorter sequence is novel and very interesting.  This introduces the benefit that many of the computations associated with each sequence can be performed in parallel.

The authors also provide theoretical arguments about the potential benefits of their approach.  However, there are multiple areas the presentation of these results should be clarified and improved.  First, it is not clear how surprising theorem 1 is: given that the derivative has a bounded Lipschitz constant is it surprising that when we integrate over shorter sequences, the Lipschitz constant of the integral, also decreases?  To this point, it would be helpful to know how tight the bound is.  If I understand the notation correctly to be big-O, this is an upper bound.  If the tightest upper bound possible is indeed exponential with T, then this result is indeed interesting, but we would need assurance (or at least a reason to conjecture) that this is indeed the case.   Second, while formal theoretical results are not needed, it would be helpful for the authors to discuss what, if anything, might be lost by breaking up longer observation sequences into shorter chunks.  For example, does this hinder the ability to learn dynamical systems with longer time scales?  Perhaps not if the network inferring initial state is accurate enough, but having an explicit discussion about this would help.   Third, the paragraph following theorem 2 appeals to it in a way that the theorem itself is not currently worded to support.  In particular, theorem 2 is currently worded as an existence proof: it is possible to find two constants such that both the state and derivative are normalized.  However, the paragraph immediately after this theorem (beginning "Hence, considering only...") appeals to this theorem as if it were a proof of necessity.  Following, the authors math shown in the proof of the theorem, I believe such a claim could be made, but the theorem would need to be re-worded (or the logic of the following paragraph made more clear).   Finally, and perhaps more importantly, I am a bit confused about how theorem 2 applies to model fitting.  In particular, I agree given a ground truth x(t) and f(t), two constants can be found such that the transformed descriptions are regularized as the authors claim.  However, when fitting models, there is flexibility in the f we chose, so that it would seem for any f and any value of 1/\tau in equation (3), can't we always find a f' = \tau f.  In other words, don't we have to restrict the flexibility of the function class we consider for f in some way for \tau to have any affect?  I acknowledge the empirical results the authors show in Fig. 5 supporting their theoretical arguments, but in those results there seems to be a large range of where RMSE of the simulation error plateaus and for which the RMS of x and f are not 1.

Finally, the authors provide empirical results comparing their method to others in two benchmark tests.  They claim to achieve state of the art results in both.  However, as the authors commendably make clear, part of their test sets are also used for validation.  They state this is fair because many of the existing methods also do the same.  However, to fully evaluate the claim that a new state of the art has been achieved, it would be helpful for the authors to show their existing results as well as results obtained when validation and test sets are fully distinct.   When comparing to previous methods which did in fact have overlapping test and validation sets, they could use the current set of results, but when comparing to methods that had disjoint test and validation sets, this second set of results could then be used to ensure a fair comparison. If their method still outperforms all existing results when comparing each on this more equal footing,  I believe they could then rigorously claim a new state of the art has been achieved.

**Summary Of The Paper:**

The authors consider the problem of learning models of continuous time dynamical systems in the presence of noise, latent variables and driving input. They present a method for doing this, which reduces the computational complexity of model fitting by breaking up longer observation sequences into shorter chunks.  They introduce the interesting idea of learning a neural network to estimate latent system state at the start of each of these chunks.   In addition, they provide theoretical results underlying their method and empirical results on benchmark applications.

**Summary Of The Review:**

Summarizing all of the above, I feel the core idea of the paper is novel and important.  However, the theoretical claims need further discussion and clarification to make their full importance more apparent.  In addition, I find it likely that the authors have indeed achieved new state of the art results on the two benchmarks they provide, but additional analyses showing how their methods work when the test and validation sets are totally disjoint are needed to confirm this.   I believe addressing these points would take an already very interesting core idea and make the paper substantially stronger.

---

> ### Author Response · Authors · 2022-11-14
> **Author Response to Reviewer mcyj part 2**
>
> **"Finally, and perhaps more importantly, I am a bit confused about how theorem 2 applies to model fitting. In particular, I agree given a ground truth x(t) and f(t), two constants can be found such that the transformed descriptions are regularized as the authors claim. However, when fitting models, there is flexibility in the f we chose, so that it would seem for any f and any value of 1/\tau in equation (3), can't we always find a f' = \tau f. In other words, don't we have to restrict the flexibility of the function class we consider for f in some way for \tau to have any affect? I acknowledge the empirical results the authors show in Fig. 5 supporting their theoretical arguments, but in those results there seems to be a large range of where RMSE of the simulation error plateaus and for which the RMS of x and f are not 1."** Indeed, in principle a rescaling of $f$ does not change the class of function that we can fit. However, proper normalization of the output of $f$ (i.e. zero mean and standard deviation of 1) is assumed in the initialization of the parameters in $f$. This initialization is an implicit restriction (aka. prior) on the function class of $f$ (see Xavier Weight Initialization for an example of this). This also answers your second question since this prior is slightly wrong (i.e. x and f are not 1) than the networks can correct this if sufficient data is available. However, if this prior is very wrong (i.e. x and f are not at all amplitude 1) then the neural networks are unable to find a good solution. I added a more clarification on this in the submission: _It is widely known that input and output normalization is essential for obtaining competitive models throughout deep learning such to respect to the prior assumptions made in for instance Xavier Weight Initialization (Glorot & Bengio, 2010)._
>
> **"Finally, the authors provide empirical results comparing their method to others in two benchmark tests. They claim to achieve state of the art results in both. However, as the authors commendably make clear, part of their test sets are also used for validation. They state this is fair because many of the existing methods also do the same. However, to fully evaluate the claim that a new state of the art has been achieved, it would be helpful for the authors to show their existing results as well as results obtained when validation and test sets are fully distinct. When comparing to previous methods which did in fact have overlapping test and validation sets, they could use the current set of results, but when comparing to methods that had disjoint test and validation sets, this second set of results could then be used to ensure a fair comparison. If their method still outperforms all existing results when comparing each on this more equal footing, I believe they could then rigorously claim a new state of the art has been achieved."** To address this concern we have added one additional benchmark where the test and validation datasets do not overlap. This benchmark is the EMPS benchmark (A. Janot, M. Gautier and M. Brunot, Data Set and Reference Models of EMPS, 2019 Workshop on Nonlinear System Identification Benchmarks, Eindhoven, The Netherlands, April 10-12, 2019.) and we added to the submission the description and results (see Figure 5 and Table 2).

---

> > ### Comment · Reviewer_mcyj · 2022-11-26
> > **Response to authors**
> >
> > I thank the authors for their detailed response to my concerns, many of which have been sufficiently addressed to warrant an increased score.

---

> ### Author Response · Authors · 2022-11-14
> **Author Response to Reviewer mcyj part 1**
>
> Thank you very much for your detailed comments, concerns and feedback.
>
> **"The authors also provide theoretical arguments about the potential benefits of their approach. However, there are multiple areas the presentation of these results should be clarified and improved. First, it is not clear how surprising theorem 1 is: given that the derivative has a bounded Lipschitz constant is it surprising that when we integrate over shorter sequences, the Lipschitz constant of the integral, also decreases?"** A bounded Lipschitz constant of $f\_\theta$ does not imply that the Lipschitz constant of the integration of $f\_\theta$ is decreasing.  This can be seen in a simple example;  Consider $\dot{x} = f\_\theta(x)/\tau = a x /\tau$ which has the Lipschitz of $L\_f = a$ (assuming positive $a$). Than considering two trajectories with different initial states yield $(x\_1(t) - x\_2(t))^2 = \exp(a t/\tau) (x\_1(0) - x\_2(0))^2$ which shows that $L\_x(t) = \exp(L\_f t/\tau)$. Hence, the Lipschitz constant of $x(t)$ can grow exponentially with time.
>
> **"To this point, it would be helpful to know how tight the bound is. If I understand the notation correctly to be big-O, this is an upper bound. If the tightest upper bound possible is indeed exponential with T, then this result is indeed interesting, but we would need assurance (or at least a reason to conjecture) that this is indeed the case."** Using the same linear example, one gets the same scaling as we obtained with the big-O notation. Hence, it does not seems that this upper bound can be lowered and hence we can consider it tight. We have included this insight in the manuscript in the Appendix.
>
> **"Second, while formal theoretical results are not needed, it would be helpful for the authors to discuss what, if anything, might be lost by breaking up longer observation sequences into shorter chunks. For example, does this hinder the ability to learn dynamical systems with longer time scales? Perhaps not if the network inferring initial state is accurate enough, but having an explicit discussion about this would help."** That is indeed the case and hence proper considerations are needed to be made while modelling. We have seen that the authors of (Beintema et al., 2021) have analysed this aspect for the discrete-time case and indeed found the effect that you mentioned. For our continuous-time extension of the approach, we have observed very similar effects, hence, we do not see it necessary to conduct an in-depth analysis of this issue. However, to clarify this, we included the following update in the paper: _"These hyperparameters are chosen based on hyperparameters analysis shown in Beintema et. al., 2021 for discrete-time. We observed similar effects of the hyperparameters for continuous-time."_
>
> **"Third, the paragraph following theorem 2 appeals to it in a way that the theorem itself is not currently worded to support. In particular, theorem 2 is currently worded as an existence proof: it is possible to find two constants such that both the state and derivative are normalized. However, the paragraph immediately after this theorem (beginning "Hence, considering only...") appeals to this theorem as if it were a proof of necessity. Following, the authors math shown in the proof of the theorem, I believe such a claim could be made, but the theorem would need to be re-worded (or the logic of the following paragraph made more clear)."** Thanks for the comment. Indeed, the inclusion of state-derivative normalization only assures the existence of a properly normalized model. That paragraph has been improved as follows; _"Hence, to assure the existence of a properly normalized model where both the state-derivative $\tilde{f}$ function and state $\tilde{x}$ are normalized, it is sufficient to include a state and state-derivative normalization factor."_

---

### Official Review · Reviewer_jmUH · 2022-11-07

**Confidence:** 4
**Correctness:** 4
**Technical Novelty And Significance:** 2
**Empirical Novelty And Significance:** 3
**Recommendation:** 5

**Clarity, Quality, Novelty And Reproducibility:**

Clarity
While the first two sections do a good job in familiarizing the reader with the subject of the study, with previous relevant work, and by stating the desired properties of the model, the manuscript lacks clarity in later sections. Particularly, it not clear how the subsection approach solves the issue of gradient-descent requiring a full forward pass. The “pipe” notation in the Proposed method section is a bit abstruse, and the role of the subsection hyperparameters na and nb should be discussed in more detail. Similarly, the role of the encoder could be examined further, for example by clarifying the relationship between the transient error, the initial state error, and the encoder loss. The manuscript also contains a few typos, the most evident being the word “reconstructability” written as “reconstrubability” in a few instances.
Additional comments on clarity:
-	"However, they only consider a fixed output function and solve their optimization problem as an optimal control problem whereas our formulation alters the simulation loss function to obtain the computationally desirable form."
This needs to be discussed further. Is it always better to alter the simulation loss? What is the "however" referring to? What is the computationally desirable form?
-	The benchmark problems are introduced very briefly. As Figure 3 is practically uninformative, it could be replaced by the equations of the underlying dynamical systems that can be easily found in other articles.
-	It is not clear in the first part of the paper that the encoder is also implemented as a neural network. One has to wait for the first line of the Results section to learn that. It would be useful to the reader to mention that earlier in the paper since the encoder is a main feature of the proposed model.

Novelty
I have concerns regarding the originality of this work compared to [Forgione & Piga 2021 and Ayed et al 2019]. Subsections and the inclusion of external inputs that render the system non-autonomous were already introduced in Forgione & Piga 2021, but do not employ an encoder to estimate the initial states. The method of Ayed 2019, instead, does incorporate an encoder to estimate the initial states but do not include inputs. By combining these methods together, the present manuscript does not appear to introduce any especially novel feature. Nevertheless, the method does improve performance and the theoretical insights estimating the effect of introducing subsections and normalization on training robustness, are surely interesting. I believe that improving the clarity of the manuscript would also emphasize the innovative aspects of the model and help the reader identifying the differences with respect of previous works.

Reproducibility
The provided jupyter notebooks do reproduce the results of the paper. However, the training implementation is contained in a python package that I could not directly inspect.


**Strength And Weaknesses:**

Strengths
The problem of inferring the dynamics underlying physical, biological, and social systems is an important one, and this manuscript introduces promising results with respect to i) the robustness of the training algorithm when the problem is broken down into overlapping shorter subsections, and ii) the intriguing possibility of introducing an encoder to infer the initial state of the system at the beginning of each subsection. Particularly interesting is the analytical proof shown in the first part of the appendix, in which they derive the temporal scaling of the Lipschitz constant in the parameters space. Also significant are the insights given on the topic of the normalization terms required to avoid pathological behavior during training. The performance of the model on the two chosen benchmarks seems impressive compared to other methods.

Weaknesses
The main claims of the paper are, in my opinion, not sufficiently explored in the results section. Specifically, a proper evaluation of the advantage in terms of computational costs compared to the other methods in Table 1 is missing. This is particularly important for other methods, like Forgione & Piga 2021 that also employ subsections. Furthermore, the authors cite Ayed et al 2019 as a similar work that also employ an encoder function, but do not compare it to their method. Along the same lines, the authors do not investigate the choice of hyperparameters such as na, nb or the length of a subsection Tdt. Exploring these hyperparameters and their impact of performance vs computing costs would greatly enhance the impact of the paper. Finally, the second benchmark only contains 500 samples in total, which seems modest if contrasted with the claims made by the authors about “accurate long-term predictions”. Perhaps a different benchmark – e.g. the EMPS dataset employed in Forgione & Piga 2021 – could be used to highlight the potential of the method, specifically the use of subsections. There’s one last aspect that I think is not particularly explored in the paper despite the initial claims: the detection of latent states. The benchmark used in the paper are not particularly well suited to evaluate the ability of the model to infer the presence of latent states. Perhaps a better benchmark could be used since this is one of the main claims of the paper.

**Summary Of The Paper:**

This conference paper aims at providing a computationally efficient algorithm to identify the underlying dynamics of nonlinear physical systems using Continuous-time (CT) neural network models that consider the presence of challenging features such as external inputs, noise in the measurement process, latent states, and robustness during training. Recent models have explored the ability of general models parametrized by CT neural networks to learn unknown nonlinear dynamics from partial observations. However, these models typically seek to include only one challenging feature – thereby ignoring the others – and are computationally intensive, especially for long time series. Building on these previous works, the authors propose to break the time series into shorter overlapping chunks, each one approximated by the same neural networks and linked by an encoder function, also approximated by a neural network. The authors convincingly show analytical evidence that this approach increases the smoothness of the cost function with respect to the parameters, and hence the robustness of the learning process, and that it can reduce the computational costs by partially parallelizing operations. The performance of the proposed model is tested on two standard benchmark problems and compared to other methods from machine learning. The novel method appears to perform on par or better than state-of-the-art models such as neural Ordinary Differential Equations (ODEs).

**Summary Of The Review:**

Summary of the review
This work addresses the important problem of reconstructing the underlying dynamics of a nonlinear system. Overall, the manuscript contains elements of interest, and the performance of the model are impressive. However, it somewhat lacks clarity and depth when discussing some key features of the model, such as the concrete advantages in computing costs, the novel aspects compared to previous works, and the advantage of utilizing the encoder to infer the initial state. Novelty is also a concern per se, since all the features of the model where already present in previous works, albeit not at simultaneously.
The authors may want to clarify the unclear aspects and expand the discussion before considering the paper for publication.

---

> ### Author Response · Authors · 2022-11-14
> **Author Response to Reviewer jmUH part 1**
>
> Thank you very much for carefully reading our paper and for your detailed comments, concerns and feedback.
>
> **"The novel method appears to perform on par or better than state-of-the-art models such as neural Ordinary Differential Equations (ODEs)."** In our opinion the proposed method performs better on the considered benchmark examples compared to the neural Ordinary Differential Equations since the RMS errors are
>
> * CCT benchmark: 0.22 CT SUBNET vs NaN neural ODE (did not converge)
> * CED benchmark: 0.115 & 0.074 CT SUBNET vs 0.141 & 0.098 neural ODE
>
> Only when we added our proposed state-derivative normalization constant to the neural ODE the results become comparable. Furthermore, SUBNET still has advantages in computational cost and training stability due to the use of subsections with encoder.
>
> **"Weaknesses The main claims of the paper are, in my opinion, not sufficiently explored in the results section. Specifically, a proper evaluation of the advantage in terms of computational costs compared to the other methods in Table 1 is missing. This is particularly important for other methods, like Forgione & Piga 2021 that also employ subsections."** Unfortunately, the authors of the publications where the indicated methods present in Table 1 come from do not report computational cost. Furthermore, those authors also did not make their codes publicly available. Hence, even if we would be rather eager to provide such a wide comparison, we are unable to do so. However, as mentioned in the text, we can show the advantages in computational cost using the big-O notation in "this optimization problem is less computationally challenging to solve if $T<N$ since the first sum can be computed in parallel". Hence, the computation costs scale as $O(T)$ for SUBNET and $O(N)$ without the use of subsections. _We added additional clarification to the submission of these scaling laws._
>
> **"Furthermore, the authors cite Ayed et al 2019 as a similar work that also employ an encoder function, but do not compare it to their method."** We think that a comparison to (Ayed et. al. 2019) would be interesting, but that would be rather challenging to realize. We would need to significantly modify the method reported in (Ayed et. al. 2019) to include (i) a parameterized output function, and (ii) external inputs and (iii) carefully consider the influence of measurement noise in the training data. Otherwise, not resolving the issues (i)-(iii) with (Ayed et. al. 2019) would not provide any meaningful results. Hence, we decided that such a comparison is not within the scope of this submission.
>
> **"Along the same lines, the authors do not investigate the choice of hyperparameters such as na, nb or the length of a subsection $T\Delta t$. Exploring these hyperparameters and their impact of performance vs computing costs would greatly enhance the impact of the paper."** The choice of hyperparameters such as $n_a$, $n_b$ or the length of a subsection $T\Delta t$ can significantly affect the performance of the learning process. The impact of these parameters has been studied by the authors of the discrete-time SUBNET approach in (Beintema et. al., 2021). In our continuous-time extension of the method, we have observed similar effects on the estimation performance as in their discrete-time case. Thus we do not see it necessary to include an in-depth analysis of this issue. However, we have included additional explanations in the paper to clarify these aspects.
>
> **"Finally, the second benchmark only contains 500 samples in total, which seems modest if contrasted with the claims made by the authors about “accurate long-term predictions”. Perhaps a different benchmark – e.g. the EMPS dataset employed in Forgione & Piga 2021 – could be used to highlight the potential of the method, specifically the use of subsections."** Thanks for the suggestion of using the EMPS benchmark. We are happy to say that the application to the EMPS has been a success. The resulting RMSE of 4.6 mm which is close to the state-of-the-art for pure back-box models. We adapted this result in the submission with a description of the EMPS benchmark and results (as seen in Figure 5 and Table 2).
>
> **"There’s one last aspect that I think is not particularly explored in the paper despite the initial claims: the detection of latent states. The benchmark used in the paper are not particularly well suited to evaluate the ability of the model to infer the presence of latent states. Perhaps a better benchmark could be used since this is one of the main claims of the paper."** As mentioned in the text, both benchmarks contain latent states since the _CCT_ has 2 states but only one water level is measured and _CED_ has 3 states but only the absolute value of the velocity of the belt is measured. Thus the encoder needs to estimate hidden states which are not included in the measurements.

---

> > ### Comment · Reviewer_jmUH · 2022-12-02
> > **Reviewer jmUH response to authors part 1**
> >
> > I want to thank the author for their careful and accurate replies.
> >
> > __"The novel method appears to perform on par or better than state-of-the-art models such as neural Ordinary Differential Equations (ODEs)." In our opinion the proposed method performs better on the considered benchmark examples compared to the neural Ordinary Differential Equations since the RMS errors are__
> > __CCT benchmark: 0.22 CT SUBNET vs NaN neural ODE (did not converge)
> > CED benchmark: 0.115 & 0.074 CT SUBNET vs 0.141 & 0.098 neural ODE
> > Only when we added our proposed state-derivative normalization constant to the neural ODE the results become comparable. Furthermore, SUBNET still has advantages in computational cost and training stability due to the use of subsections with encoder.__
> >
> > I understand the authors point, but I find it a bit unfair to compare their model to a model that, in the hands of the authors, did not converge. Furthermore, the additional benchmark added by author after my and another reviewer's suggestion shows that the proposed method's performance falls short of another black box method (Weigand et al. 2021). Therefore, I think it would be fair to state in the paper that the method can slightly outperforms or underperform other state-of the art models depending on the benchmark task.
> >
> > __"Weaknesses The main claims of the paper are, in my opinion, not sufficiently explored in the results section. Specifically, a proper evaluation of the advantage in terms of computational costs compared to the other methods in Table 1 is missing. This is particularly important for other methods, like Forgione & Piga 2021 that also employ subsections." Unfortunately, the authors of the publications where the indicated methods present in Table 1 come from do not report computational cost. Furthermore, those authors also did not make their codes publicly available. Hence, even if we would be rather eager to provide such a wide comparison, we are unable to do so. However, as mentioned in the text, we can show the advantages in computational cost using the big-O notation in "this optimization problem is less computationally challenging to solve if  since the first sum can be computed in parallel". Hence, the computation costs scale as  for SUBNET and  without the use of subsections. We added additional clarification to the submission of these scaling laws.__
> >
> > I thank the authors for their clarifying response. I understand the computational advantage obtained by parallelizing the computations across different subsections. Yet Forgione & Piga 2021 also employ subsections but do not use an encoder function. What would be the computational cost compared to their work?
> >
> > __"Furthermore, the authors cite Ayed et al 2019 as a similar work that also employ an encoder function, but do not compare it to their method." We think that a comparison to (Ayed et. al. 2019) would be interesting, but that would be rather challenging to realize. We would need to significantly modify the method reported in (Ayed et. al. 2019) to include (i) a parameterized output function, and (ii) external inputs and (iii) carefully consider the influence of measurement noise in the training data. Otherwise, not resolving the issues (i)-(iii) with (Ayed et. al. 2019) would not provide any meaningful results. Hence, we decided that such a comparison is not within the scope of this submission.__
> >
> > I understand the authors point, however I am not I understand points i) and iii) in their response. It would be great if they could clarify what they mean by "including a parameterized output function", and "carefully consider the influence of measurement noise in the training data". As for point ii), it could be overcome by comparing the models over a dataset that doesn't require external inputs.
> >
> > __"Along the same lines, the authors do not investigate the choice of hyperparameters such as na, nb or the length of a subsection . Exploring these hyperparameters and their impact of performance vs computing costs would greatly enhance the impact of the paper." The choice of hyperparameters such as ,  or the length of a subsection  can significantly affect the performance of the learning process. The impact of these parameters has been studied by the authors of the discrete-time SUBNET approach in (Beintema et. al., 2021). In our continuous-time extension of the method, we have observed similar effects on the estimation performance as in their discrete-time case. Thus we do not see it necessary to include an in-depth analysis of this issue. However, we have included additional explanations in the paper to clarify these aspects.__
> >
> > Could the authors please further clarify the concrete difference between their approach and that of Beintema et al. 2021?

---

> > > ### Author Response · Authors · 2022-12-08
> > > **Author response to reviewer response jmUH part 1**
> > >
> > >
> > > Thank you for your further comments and feedback on the new version of the paper and our response. We would like to address the last of your concerns.
> > >
> > > **"I understand the authors point, but I find it a bit unfair to compare their model to a model that, in the hands of the authors, did not converge. Furthermore, the additional benchmark added by author after my and another reviewer's suggestion shows that the proposed method's performance falls short of another black box method (Weigand et al. 2021). Therefore, I think it would be fair to state in the paper that the method can slightly outperforms or underperform other state-of the art models depending on the benchmark task."**
> > >
> > > In our opinion, the comparison to NeuralODE is a fair comparison since, *(i)* the NeuralODE is developed for estimating continuous-time systems so our use case is within this setting, *(ii)* we use the implementation provided by NeuralODE only adding external inputs. Thus the non-converge is a fact of the NeuralODE method for the considered benchmarks.
> > >
> > > For your second point, as mentioned in the text, the method (Weigand et al. 2021) includes additional prior information on the system by enforcing input-output stability which is very important for the EMPS benchmark. For instance, on the CCT benchmark where input-output stability is less important Weigand et al. 2021 performed at 0.39 RMS (which will be added to Table 1 in the final revision.) whereas our method has 0.22 RMS. Hence, the proposed method is able to achieve state-of-the-art performance on a wide range of benchmarks, only being beaten by methods which are more tuned to specific benchmarks.
> > >
> > > **"I thank the authors for their clarifying response. I understand the computational advantage obtained by parallelizing the computations across different subsections. Yet Forgione & Piga 2021 also employ subsections but do not use an encoder function. What would be the computational cost compared to their work?"**
> > >
> > > The computational cost of Forgione & Piga 2021 scales as $\mathcal{O}(T)$ which is of course the same scaling as we present. However, since Forgione & Piga 2021 uses parameterized initial states in their subsections, the number of initial states which should be estimated grows linearly with the number of subsections. Hence, the model complexity grows as  $\mathcal{O}(N)$ whereas the proposed method has a constant model complexity i.e. $\mathcal{O}(1)$ since the encoder is independent of the dataset size.
> > >
> > > **"I understand the authors point, however I am not I understand points i) and iii) in their response. It would be great if they could clarify what they mean by "including a parameterized output function", and "carefully consider the influence of measurement noise in the training data". As for point ii), it could be overcome by comparing the models over a dataset that doesn't require external inputs."**
> > >
> > > By "including a parameterized output function" we mean that we include a non-fixed output function that maps the states to a (measured) output signal. This output function $h$ is parameterized by, for instance, a neural network. Secondly, "carefully consider the influence of measurement noise in the training data" means that the application of the method on a benchmark where the output and/or dynamics are subject to disturbing noise requires additional considerations. In case this is ignored, it can introduce a bias or higher variance on the obtained model estimate. As you say, we could also compare on datasets without external inputs. However, that is not the focus of our work. Such an in-depth comparison is left for future research.
> > >
> > > **"Could the authors please further clarify the concrete difference between their approach and that of Beintema et al. 2021?"**
> > >
> > > The authors of Beintema et al. 2021 considered the discrete-time modeling case whereas we consider the continuous-time modeling case. Moving from a discrete-time to a continuous-time setting is non-trivial, and has an impact on the construction/existence of an encoder function, the cost smoothness analysis, the simulation of the model during training, the impact of noise on the system, and it also required additional normalization considerations (normalizing both states and state-derivatives).

---

> ### Author Response · Authors · 2022-11-14
> **Author Response to Reviewer jmUH part 2**
>
> **"Particularly, it not clear how the subsection approach solves the issue of gradient-descent requiring a full forward pass."** It does not require a full forward pass since each subsection is completely independent of one another. For example, consider the computation of $\hat{y}\_{6|3}$ and $\hat{y}\_{7|4}$. One can write with the relation given in (3) it as follows;
>
> $
> \hat{y}\_{6|3} = h\_\theta(x\_{6|3})
> $
>
> $
> x\_{6|3} = \text{ODEsolve} [ \frac{1}{\tau} f\_\theta, x\_{5|3}, u\_{5}, \Delta t ]
> $
>
> $
> x\_{5|3} = \text{ODEsolve} [ \frac{1}{\tau} f\_\theta, x\_{4|3}, u\_{4}, \Delta t ]
> $
>
> $
> x\_{4|3} = \text{ODEsolve} [ \frac{1}{\tau} f\_\theta, x\_{3|3}, u\_{3}, \Delta t ]
> $
>
> $
> x\_{3|3} = \psi\_\theta(u\_{3-1},...,u\_{3-n\_b}, y\_{3-1}, ..., y\_{3-n\_a})
> $
>
> and
>
> $
> \hat{y}\_{7|4} = h\_\theta(x\_{7|4})
> $
>
> $
> x\_{7|4} = \text{ODEsolve} [ \frac{1}{\tau} f\_\theta, x\_{6|4}, u\_{6}, \Delta t ]
> $
>
> $
> x\_{6|4} = \text{ODEsolve} [ \frac{1}{\tau} f\_\theta, x\_{5|4}, u\_{5}, \Delta t ]
> $
>
> $
> x\_{5|4} = \text{ODEsolve} [ \frac{1}{\tau} f\_\theta, x\_{4|4}, u\_{4}, \Delta t ]
> $
>
> $
> x\_{4|4} = \psi\_\theta(u\_{4-1},...,u\_{4-n\_b}, y\_{4-1}, ..., y\_{4-n\_a})
> $
>
>
> which shows that the computation graph of $\hat{y}\_{6|3}$ is completely independent of $\hat{y}\_{7|4}$ and hence they can be computed on different CPU threads. Furthermore, in both examples, the state only has to be propagated 3 times after the use of the encoder. Hence, this does not require starting from index 1 and thus no full forward pass is required.
>
> **"The “pipe” notation in the Proposed method section is a bit abstruse."** The pipe notation is similar to the notation of Kalman filtering and conditional probability distributions. We added additional clarifications to the text of this relationship. Added to submission: _This pipe notation is similar to the notation used in Kalman filtering and conditional probability distributions (Chui et al., 2017)._
>
> **"Similarly, the role of the encoder could be examined further, for example by clarifying the relationship between the transient error, the initial state error, and the encoder loss."** If the encoder does not construct the correct state then we have an initial state error. This initial state-error results in a residual with the output trajectories with the correct initial state. This residual decays with time for strictly stable systems, hence, this can be interpreted as a transient error. Lastly, since the encoder loss sums over multiple small subsections which can include the transient and thus the encoder loss includes the transient error.
>
> **"The manuscript also contains a few typos, the most evident being the word “reconstructability” written as “reconstrubability” in a few instances."** Thank you for notifying us. We corrected the error and also multiple other typos.
>
> **"Additional comments on clarity: - "However, they only consider a fixed output function and solve their optimization problem as an optimal control problem whereas our formulation alters the simulation loss function to obtain the computationally desirable form." This needs to be discussed further. Is it always better to alter the simulation loss? What is the "however" referring to? What is the computationally desirable form?"** In that sentence we do not argue that our method is better only that it is different. For our case, the computationally desirable form refers to a form that can be solved with the conventual gradient descent method in a stable and computationally efficient manner. We added additional clarification to the sentence.
>
> **"The benchmark problems are introduced very briefly. As Figure 3 is practically uninformative, it could be replaced by the equations of the underlying dynamical systems that can be easily found in other articles."** We believe that the statement that Figure 3 is uninformative, is not entirely valid. This is a photo and a schematic representation of the systems on which these measurements are obtained. For example, it provides visual information on how the two-tank system setup is realized. It also emphasizes that the data is obtained from real-life systems and not from numerical simulations. As it are physical systems that provide the data, the exact underlying system equations are unknown. Only approximate equations can be provided. However, that would be somewhat misleading as they present a simplification of the true modeling challenge.
>
> **"It is not clear in the first part of the paper that the encoder is also implemented as a neural network. One has to wait for the first line of the Results section to learn that. It would be useful to the reader to mention that earlier in the paper since the encoder is a main feature of the proposed model."** Thank you for pointing this out. We clarified this point in the manuscript where the encoder is introduced.

---

> > ### Comment · Reviewer_jmUH · 2022-12-02
> > **Reviewer jmUH response to authors part 2**
> >
> > __"Particularly, it not clear how the subsection approach solves the issue of gradient-descent requiring a full forward pass." It does not require a full forward pass since each subsection is completely independent of one another. For example, consider the computation of [...]
> >  and [...]. One can write with the relation given in (3) it as follows; [...] and [...] which shows that the computation graph of is completely independent of [...] and hence they can be computed on different CPU threads. Furthermore, in both examples, the state only has to be propagated 3 times after the use of the encoder. Hence, this does not require starting from index 1 and thus no full forward pass is required.__
> >
> > I apologize, I had misunderstood what the authors mean by full forward pass, I now understand thanks tot he authors' response.
> >
> > __"The “pipe” notation in the Proposed method section is a bit abstruse." The pipe notation is similar to the notation of Kalman filtering and conditional probability distributions. We added additional clarifications to the text of this relationship. Added to submission: This pipe notation is similar to the notation used in Kalman filtering and conditional probability distributions (Chui et al., 2017).__
> >
> > Thanks for the clarification.
> >
> > __"Similarly, the role of the encoder could be examined further, for example by clarifying the relationship between the transient error, the initial state error, and the encoder loss." If the encoder does not construct the correct state then we have an initial state error. This initial state-error results in a residual with the output trajectories with the correct initial state. This residual decays with time for strictly stable systems, hence, this can be interpreted as a transient error. Lastly, since the encoder loss sums over multiple small subsections which can include the transient and thus the encoder loss includes the transient error.__
> >
> > I am thankful to the authors for the explanation. I guess what I meant was more towards a quantification of the contribution of the initial state error determined by the encoder compared to models that do not employ and encoder. This way the specific contribution of the encoder to the performance could be quantified.
> >
> > __"Additional comments on clarity: - "However, they only consider a fixed output function and solve their optimization problem as an optimal control problem whereas our formulation alters the simulation loss function to obtain the computationally desirable form." This needs to be discussed further. Is it always better to alter the simulation loss? What is the "however" referring to? What is the computationally desirable form?" In that sentence we do not argue that our method is better only that it is different. For our case, the computationally desirable form refers to a form that can be solved with the conventual gradient descent method in a stable and computationally efficient manner. We added additional clarification to the sentence.__
> >
> > Thanks for the addition.
> >
> > __"The benchmark problems are introduced very briefly. As Figure 3 is practically uninformative, it could be replaced by the equations of the underlying dynamical systems that can be easily found in other articles." We believe that the statement that Figure 3 is uninformative, is not entirely valid. This is a photo and a schematic representation of the systems on which these measurements are obtained. For example, it provides visual information on how the two-tank system setup is realized. It also emphasizes that the data is obtained from real-life systems and not from numerical simulations. As it are physical systems that provide the data, the exact underlying system equations are unknown. Only approximate equations can be provided. However, that would be somewhat misleading as they present a simplification of the true modeling challenge.__
> >
> > I understand the authors' point regarding the absence of exact equations. I still think the picture is rather uninformative and could be replaced by a schematic illustration.
> >
> > __"It is not clear in the first part of the paper that the encoder is also implemented as a neural network. One has to wait for the first line of the Results section to learn that. It would be useful to the reader to mention that earlier in the paper since the encoder is a main feature of the proposed model." Thank you for pointing this out. We clarified this point in the manuscript where the encoder is introduced.__
> >
> > Thank you for clarifying.

---

> > > ### Author Response · Authors · 2022-12-08
> > > **Author response to reviewer response jmUH part 2**
> > >
> > > **"I am thankful to the authors for the explanation. I guess what I meant was more towards a quantification of the contribution of the initial state error determined by the encoder compared to models that do not employ and encoder. This way the specific contribution of the encoder to the performance could be quantified."**
> > >
> > > We agree a more in-depth analysis of the influence of the encoder and initial state error on the performance is an interesting topic for future research when it comes to the specific nature of the encoder training. However, the impact of initial states has been studied in detail in system theory publications and the identification literature (e.g. multiple shooting contributions).

---

> ### Author Response · Authors · 2022-11-14
> **Author Response to Reviewer jmUH part 3**
>
> **"Novelty I have concerns regarding the originality of this work compared to [Forgione & Piga 2021 and Ayed et al 2019]. Subsections and the inclusion of external inputs that render the system non-autonomous were already introduced in Forgione & Piga 2021, but do not employ an encoder to estimate the initial states. The method of Ayed 2019, instead, does incorporate an encoder to estimate the initial states but do not include inputs. By combining these methods together, the present manuscript does not appear to introduce any especially novel feature."** We would like to humbly argue with the statements of the reviewer: The aim of this submission is to introduce a method which is robust and theoretically well-founded. The fact that parts of the method build on elements from existing work does not diminish its novelty since we _(i)_ extend these methods to the general case of latent states, external inputs, measurement noise and long-sequence modelling and _(ii)_ provide new insight in these methods using theoretical results like Theorem 1 on subsections and Lipschitz analysis and necessary conditions for the existence of the encoder. Lastly, we also provide novel elements such as state-derivative normalization and apply our method to data obtained from real systems to gain practical insight into the method.
>
> **"Reproducibility The provided jupyter notebooks do reproduce the results of the paper. However, the training implementation is contained in a python package that I could not directly inspect."** We have added additional information in the `readme.md` to show how to install the required packages and (re-)run the notebooks. We also provided the notebook for the EMPS training and results.
>
> Thanks again for all the feedback and comments.

---

> > ### Comment · Reviewer_jmUH · 2022-12-02
> > **Reviewer jmUH response to authors part 3**
> >
> > __"Novelty I have concerns regarding the originality of this work compared to [Forgione & Piga 2021 and Ayed et al 2019]. Subsections and the inclusion of external inputs that render the system non-autonomous were already introduced in Forgione & Piga 2021, but do not employ an encoder to estimate the initial states. The method of Ayed 2019, instead, does incorporate an encoder to estimate the initial states but do not include inputs. By combining these methods together, the present manuscript does not appear to introduce any especially novel feature." We would like to humbly argue with the statements of the reviewer: The aim of this submission is to introduce a method which is robust and theoretically well-founded. The fact that parts of the method build on elements from existing work does not diminish its novelty since we (i) extend these methods to the general case of latent states, external inputs, measurement noise and long-sequence modelling and (ii) provide new insight in these methods using theoretical results like Theorem 1 on subsections and Lipschitz analysis and necessary conditions for the existence of the encoder. Lastly, we also provide novel elements such as state-derivative normalization and apply our method to data obtained from real systems to gain practical insight into the method.__
> >
> > I agree with the authors that Theorem 1, the Lipschitz analysis and the state-derivative normalization are novel analysis aspects. I will increase the manuscript score since the authors addressed most of my concerns.
> >
> > __"Reproducibility The provided jupyter notebooks do reproduce the results of the paper. However, the training implementation is contained in a python package that I could not directly inspect." We have added additional information in the readme.md to show how to install the required packages and (re-)run the notebooks. We also provided the notebook for the EMPS training and results.__
> >
> > I want to thank the authors for the additional information and notebook.

---

> > > ### Author Response · Authors · 2022-12-08
> > > **Author response to reviewer response jmUH part 3**
> > >
> > > **"I agree with the authors that Theorem 1, the Lipschitz analysis and the state-derivative normalization are novel analysis aspects. I will increase the manuscript score since the authors addressed most of my concerns."**
> > >
> > > Thank you very much for considering an increase in the manuscript score. Your detailed comments and feedback have helped us a lot in improving the manuscript.

---

### Author Response · Authors · 2022-11-14
**Additional Benchmark Result Added**

Upon the request of two reviewers, we added one additional benchmark and associated results of the CT SUBNET approach to the submission. This benchmark is the Electro-Mechanical Positioning System (EMPS) (A. Janot, M. Gautier and M. Brunot, Data Set and Reference Models of EMPS, 2019 Workshop on Nonlinear System Identification Benchmarks, Eindhoven, The Netherlands, April 10-12, 2019.).

---

> ### Comment · Reviewer_jmUH · 2022-11-23
> **Additional EMPS benchmark**
>
> I want to thank the authors for including this benchmark I suggested.

---

### Decision · Program_Chairs · 2023-01-20

**Decision:**

Accept: poster

**Justification For Why Not Higher Score:**

While the reviewers found the method to be novel, they thought more could be done to improve the clarity of the paper and in the evaluation of the method.

**Justification For Why Not Lower Score:**

The proposed method is clearly novel and holds promise.

**Metareview: Summary, Strengths And Weaknesses:**

The paper proposes a new method for continuous time state space modeling. It has two components: a method for training on short subsequences of the training set which is shown to improve the smoothness of the cost function; and a new normalization strategy which improves the conditioning of the model. The method is shown to improve upon SOTA, especially in computational complexity and reliability, and theoretical guarantees of the subset training procedure are also provided.

**Note From Pc:**

if the above contains the word "oral" or "spotlight" please see: "oral" presentation means -> notable-top-5% and "spotlight" means -> notable-top-25%. As stated in our emails, we are disassociating presentation type from AC recommendations